# Progress Reward Model for Reinforcement Learning via Large Language Models

**Xiuhui Zhang,  Ning Gao,  Xingyu Jiang,**
**Yihui Chen,  Yuheng Pan,  Mohan Zhang,  Yue Deng**[*]

Beihang University
37 Xueyuan Road, Haidian District, Beijing
{zhangxiuhui, gaoning_ai, jxy33zrhd}@buaa.edu.cn
{21375399, fzxl2003, zmh666, ydeng}@buaa.edu.cn

## Abstract

Traditional reinforcement learning (RL) algorithms face significant limitations in handling long-term tasks with sparse rewards. Recent advancements have leveraged large language models (LLMs) to enhance RL by utilizing their world knowledge for task planning and reward generation. However, planning-based approaches often depend on pre-defined skill libraries and fail to optimize low-level control policies, while reward-based methods require extensive human feedback or exhaustive searching due to the complexity of tasks. In this paper, we propose the Progress Reward Model for RL (PRM4RL), a novel framework that integrates task planning and dense reward to enhance RL. For high-level planning, a complex task is decomposed into a series of simple manageable subtasks, with a subtask-oriented, fine-grained progress function designed to monitor task execution progress. For low-level reward generation, inspired by potential-based reward shaping, we use the progress function to construct a Progress Reward Model (PRM), providing theoretically grounded optimality and convergence guarantees, thereby enabling effective policy optimization. Experimental results on robotics control tasks demonstrate that our approach outperforms both LLM-based planning and reward methods, achieving state-of-the-art performance. The code is available at `https://github.com/deng-ai-lab/PRM4RL`

## 1 Introduction

Reinforcement learning (RL) has shown exceptional potential in solving sequential decision-making problems. However, in real-world scenarios involving complex, long-term tasks with sparse rewards, agents face significant challenges in planning and decision-making due to insufficient feedback signals[1, 2]. Traditional RL frameworks address these challenges primarily by focusing on two strategies: task decomposition and reward shaping. Hierarchical RL[3, 4] simplifies decision-making by autonomously decomposing complex tasks into hierarchical subtasks, while reward shaping techniques provide dense training signals through the reconstruction of reward functions, such as inverse RL[5, 6, 7] and intrinsic reward mechanisms[8]. Despite their advantages, both approaches heavily rely on expert data for priors[9] or supervision[10], and the large-scale acquisition of such data is often challenging, limiting the scalability and applicability of these methods.

Recent advancements in large language model (LLM) research have made significant progress in areas such as dialogue[11, 12], sequential decision-making[13, 14], and coding[15, 16, 17]. By leveraging the extensive and diverse human knowledge acquired during the pre-training phase, LLMs

---

[*]Corresponding author

39th Conference on Neural Information Processing Systems (NeurIPS 2025).

possess a broad range of world knowledge and emergent reasoning abilities[18, 19]. Consequently, LLM-Augmented RL has emerged as a promising approach to overcome the challenges faced by traditional RL methods[20]. To tackle the issues of long-term planning and sparse rewards, previous LLM-based approaches have evolved into two main paradigms: LLM as a planner[21, 22, 23] and LLM as a rewarder[24, 25, 26], as illustrated in Figure 1. As a planner, LLM decomposes complex, long-term tasks into a series of short-term, goal-oriented subtasks, incorporating subtask priors into the decision-making process. As a rewarder, LLM constructs a dense reward function for sparse-reward tasks, providing guidance to the underlying policy based on the meaningful priors encoded within the LLM.

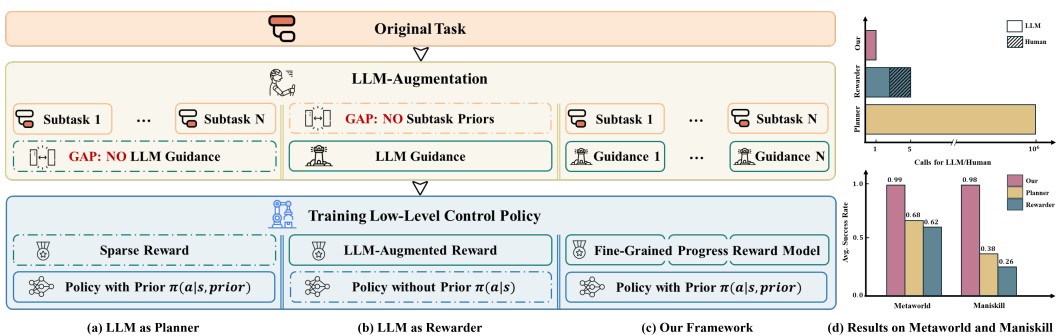

Figure 1: Quick view. (a), (b) and (c) shows the structual similarities and differences between LLM-Augmented methods. (d) is the average evaluation results on Metaworld[27] and Maniskill[28].

However, LLM as a planner primarily focuses on high-level planning. In the absence of LLM-augmented dense reward guidance, task decomposition alone—relying solely on natural-language task priors—is insufficient to support the effective learning of subtasks[29, 22]. Previous approaches have alleviated this limitation by constructing pre-trained policy libraries[21] or incorporating traditional motion planning to link subtasks[22], yet these methods lack flexibility and efficiency due to their non-end-to-end nature. On the other hand, LLM as a rewarder provides low-level guidance. However, without subtask decomposition, constructing a reward function for an entire long-term, complex task remains a significant challenge for LLMs[30, 31]. Previous methods have enhanced the quality of LLM-generated reward functions by incorporating human feedback[24] or evolutionary search algorithms[25, 26], but these approaches still require extensive search and significant human expert assistance. As shown in Figures 1a and 1b, both of these approaches focus on a single aspect of RL, neglecting the importance of integrating both high-level planning and low-level reward shaping.

To address the challenges of long-term planning and sparse rewards in reinforcement learning more efficiently, we propose a novel framework, Progress Reward Model for Reinforcement Learning (PRM4RL), which integrates both high-level long-term planning and low-level dense rewards to provide a comprehensive solution, as illustrated in Figure 1c. In high-level planning, the task is decomposed into subtasks, each with termination conditions and fine-grained progress evaluations. For low-level reward generation, we utilize the progress to construct an efficient dense reward signal with theoretically grounded optimality and convergence guarantees. In practice, our pipeline decomposes a complex task into a series of manageable, goal-oriented subtasks, each associated with a natural language subtask prior. For each subtask, we build a fine-grained progress function that dynamically tracks the execution of the task. Inspired by the potential field theorem, we regard the progress function as a potential function and design a Progress Reward Model with optimality and convergence guarantees. The evaluation results in Metaworld[27] and Maniskill[28] as shown in Figure 1d demonstrate the effectiveness of our proposed framework, with significant improvements in success rate and reduced LLM calls, outperforming existing methods in both performance and efficiency.

## 2 Related Works

### 2.1 LLM for Planning

Large language models (LLMs), enriched with diverse world knowledge and human priors during their pretraining phase, possess the ability to understanding and reasoning[18, 19, 17, 16, 11]. When provided with relevant task information, LLMs are able to decompose complex, long-term tasks into simpler, short-term tasks, each focused on specific objectives[32, 33]. Numerous approaches have leveraged Task and Motion Planning (TAMP) to realize such task decomposition in robotics scenes[34, 35]. However, when using RL for low-level control strategies, a significant gap arises between high-level planning and low-level control—how to efficiently leverage the natural language plans generated by LLMs to guide policy training? SayCan[21] addresses this challenge by utilizing a set of pretrained policies, where the LLM simply selects the appropriate policy based on the plans. Plan-Seq-Learn[22] uses traditional motion planning to link subtasks, alleviating the burden on RL policies. In text-based environments, ELLM[23] trains RL policies by describing state transition in natural language and computing the similarity between the executed actions and the plan, using this similarity as a reward signal. However, this approach is computationally expensive and not suitable for robotics tasks. In contrast, our method constructs an efficient and dense reward signal through the Progress Reward Model (PRM), effectively bridging the gap between high-level planning and low-level policy training.

### 2.2 LLM for Reward Generation

Reward functions are the primary mechanism to guide RL training[36, 37], leading to numerous approaches that utilize LLMs to generate executable reward functions that provide dense rewards. Text2Reward[24] uses LLMs to generate executable python functions based on a goal described in natural language, with human involvement in the loop. Eureka[25] extends this paradigm by introducing evolutionary search and automatic reward refinement through LLMs. CurricuLLM[38] translates natural language description of tasks in executable task code, including the reward code and goal distribution code. REvolve[26] translates implicit human knowledge into explicit reward functions for training RL with human feedback. However, these methods treat the task as a whole and do not account for the potential stage-wise features of complex tasks. As a result, the generation process of reward function tend to be highly complex and often require human feedback[24, 26] or extensive search[25]. In contrast, our approach simplifies the reward generation process by decomposing the task into short-term, simple subtasks, providing more specific and clear guidance for each stage of the task.

## 3 Preliminary

In standard reinforcement learning (RL)[1] tasks, the problem is typically modeled as a Markov Decision Process (MDP), denoted by $\mathcal{M} = \langle \mathcal{S}, \mathcal{A}, \mathcal{T}, R, \gamma, \rho_0 \rangle$. Here, $\mathcal{S}$ and $\mathcal{A}$ represent the state and action spaces, respectively, $\gamma$ is the discount factor, and $\rho_0$ is the distribution of initial states. The dynamics of environment are captured by the transition function $\mathcal{T} : \mathcal{S} \times \mathcal{A} \times \mathcal{S} \to [0, 1]$, which defines the probability of transitioning from one state to another given a specific action. The reward function $R : \mathcal{S} \times \mathcal{A} \to \mathbb{R}$ quantifies the immediate reward received for taking an action in a particular state. A trajectory $\tau = (s_0, a_0, r_0, \ldots, s_T, a_T, r_T)$ describes the sequence of states, actions, and rewards encountered by an agent following a decision-making policy $\pi$. At each time step $t$, the agent selects an action $a_t$ according to $a_t \sim \pi(\cdot|s_t)$, and the environment responds by transitioning to the next state $s_{t+1} \sim \mathcal{T}(\cdot|s_t, a_t)$. The state-action value function (Q-function) $Q^\pi(s, a)$ represents the expected cumulative return when taking action $a$ in state $s$ and subsequently following policy $\pi$:

$$Q^\pi(s, a) = \mathbb{E}_{\tau \sim p(\tau|\pi)} \left[ \sum_{k=0}^{T} \gamma^k R(s_{t+k}, a_{t+k}) \,\middle|\, s_t = s, a_t = a \right] \tag{1}$$

The objective in RL is for the agent to learn a policy $\pi : \mathcal{S} \to \mathcal{A}$ that maximizes the expected cumulative return defined as:

$$J(\pi) = \mathbb{E}_{\tau \sim p(\tau|\pi)} \left[ \sum_{t=0}^{T} \gamma^t R(s_t, a_t) \right] \tag{2}$$

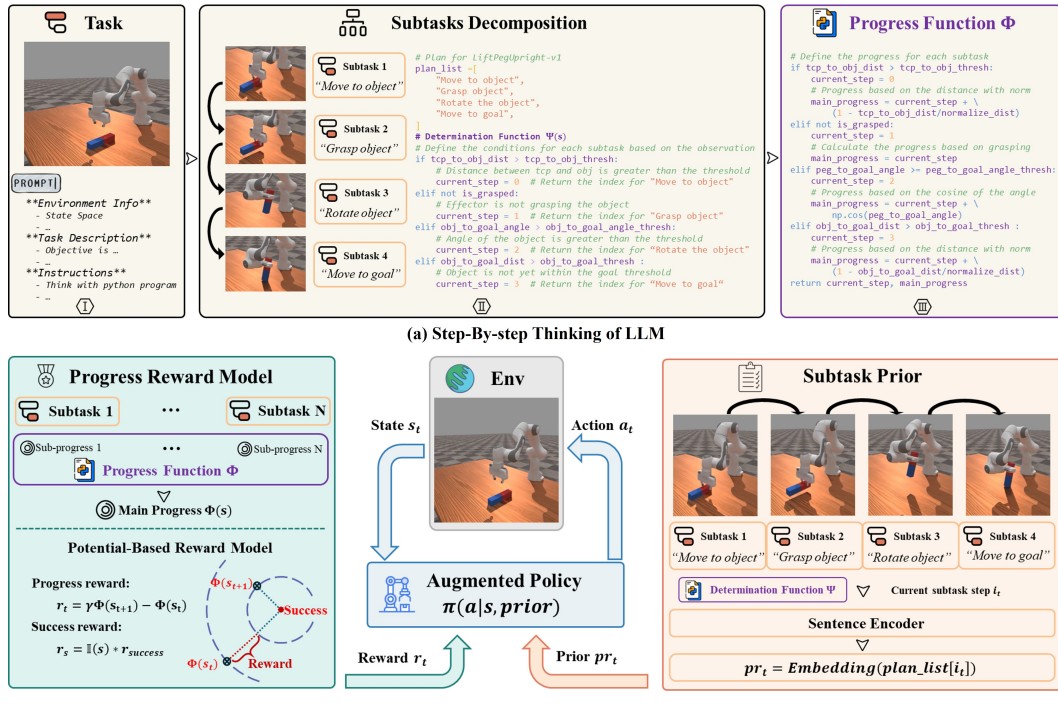

Figure 2: An overview of our framework. (a) We prompt the LLM using a Chain-of-Thought approach in Python code format. The reasoning process is provided as comments (e.g., using #), while the intermediate steps and final response are represented as executable Python statements. (b) Our framework augments RL in two ways: subtask prior and the PRM rewards.

## 4    Method

Our framework, as illustrated in Figure 2, bridges the gap between high-level planning and low-level training. We begin by decomposing a complex long-term task into a series of short-term subtasks, with a determination function Ψ to classify the current subtask (Section 4.1). Next, we construct a potential-based Progress Reward Model (PRM) derived from the progress function Φ, which generates effective reward signals that are proved to have convergence and optimality guarantees (Section 4.2). Finally, the policy is augmented using both the PRM reward and the subtask priors (Section 4.3).

**LLM usage pattern.** In our framework, we prompt an LLM to decompose the tasks and generate the determination function Ψ and progress function Φ. Considering that the intermediate inference steps and the required final response are primarily represented as Python code (as shown later in Section 4.1, 4.2), inspired from PAL[39], we require the LLM to generate programs as intermediate steps, as shown in Figure 2a. Specifically, we instruct the LLM to generate natural language reasoning steps using comment syntax (e.g., '#' in Python), while the code itself is generated normally. Through this program-aided Chain-of-Thought reasoning, the LLM can generate both the plan and progress function in a single interaction.

### 4.1    Subtask Decomposition

Decomposing a long-term task into a series of short-term, manageable tasks provides the policy with subtask priors and alleviates the complexity of constructing the progress function. Existing methods[21, 22] have demonstrated that, given sufficient environment-related information, LLMs are capable of task decomposition. In our framework, we leverage an LLM to break down the task into a series of subtasks, each contributing to the success of the overall task. We construct a Pythonic representation prompt that includes semantically rich contextual information such as state

---
**Algorithm 1** PRM4RL
---
**Require:** pythonic LLM prompt $prompt$, sentence encoder $\mathbf{E}$, policy $\pi_\theta$, MDP $\mathcal{M} = \langle \mathcal{S}, \mathcal{A}, \mathcal{T}, R, \gamma, \rho_0 \rangle$, RL optimization algorithm $Optimize\_Algo$
// Request LLM
$plan\_list, \Psi(s), \Phi(s) = LLM(prompt)$
// Augment State Space
$\mathcal{P} : prior = \mathbf{E}(plan\_list[\Psi(s)])$
// Augment Reward Function
$R^{PRM} = \gamma \cdot \Phi(s_{t+1}) - \Phi(s_t) + \mathbb{I}(s_{t+1}) \cdot r_{bonus}$
// Optimizing policy under augmented MDP
$\mathcal{M}' = \langle \mathcal{S} + \mathcal{P}, \mathcal{A}, \mathcal{T}, R^{PRM}, \gamma, \rho_0 \rangle$
$\theta^* = Optimize\_Algo(M')$
---

space, environmental descriptions, and task descriptions (detailed in Appendix A.5), providing a high-dimensional abstraction of environment-specific information.

While decomposing the subtasks, the LLM is also required to generate a natural language form *subtask prior*, which is a description of the corresponding subtask. To ensure consistency and generalization across the robotics control domain, we designed a fixed pattern for the subtask priors. Specifically, we instruct the LLM to output each subtask prior in a "verb + noun" format (e.g., 'Move to object' or 'Grasp object'), significantly enhancing the generalization ability of the trained policy. The expected output from the LLM is a Python list, called *plan_list*, as shown in Figure 2a(II), where each element corresponds to a subtask prior.

However, guiding the policy with subtask priors presents another challenge: real-time tracking of which subtask the policy is currently executing. Previous approaches require calling the LLM at each timestep, either to regenerate a real-time plan[23] or to check the completion of ongoing subtasks[22], leading to significant time and computational overhead.

Our key insight is that, with a thorough understanding of the state space, we can determine which subtask the policy is in based solely on current state. For the previously generated *plan_list*, we instruct the LLM to identify the completion condition for each subtask and locate the relevant features in the state space. We then construct a subtask determination function $\Psi(s)$, where given state $s$, it will return the index of current subtask in *plan_list*. As shown in Figure 2a(II), the LLM ultimately generates an efficient, low-cost logical chain that ensures rigorous subtask determination.

## 4.2 Potential-Based Progress Reward Model

Our framework aims to construct a dense and efficient reward signal to guide the training of low-level policies. While previous approaches[24, 25, 26] directly design a reward function, we choose to model the progress of task execution instead. This approach is more task-relevant and provides stronger theoretical guarantees on optimality and convergence as proved later.

With the decomposition of subtasks, the focus of our progress function shifts from a long-term, complex task to short-term, simpler subtasks. This transformation simplifies task comprehension and the design of usable metrics. Based on the established subtask determination logic $\Psi$, we instruct the LLM to further define a progress function for each subtask, and then integrating these fine-grained sub-progress indicators into an overall evaluation of the task progress $\Phi(s)$, as shown in Figure 2a(III). The progress function offers a precise and efficient method for monitoring overall task execution.

Inspired by potential-based framework, we treat the progress function $\Phi(s)$ as a form of potential and construct the Progress Reward Model (PRM) using the temporal difference form:

$$R_t^{PRM} = \gamma \cdot \Phi(s_{t+1}) - \Phi(s_t) + \mathbb{I}(s_{t+1}) \cdot r_{bonus} \tag{3}$$

where $R_t^{PRM}$ is the dense reward given by PRM at timestep $t$, $\gamma$ is the discount factor, and the indicator function $\mathbb{I}$ equals 1 when $s_{t+1}$ meets the success conditions and 0 otherwise. $\Phi(s_{t+1})$ and $\Phi(s_t)$ represent the progress at timesteps $t+1$ and $t$, respectively. The PRM reward can be regarded as consisting of two parts: a sparse success reward and a potential-based reward shaping term. Following [40], we present the following theorem:

**Theorem 4.1** (Policy Invariance). *Define an MDP $M = \langle \mathcal{S}, \mathcal{A}, \mathcal{T}, R, \gamma, \rho_0 \rangle$, where $R_t = \mathbb{I}(s_{t+1}) \cdot r_{bonus}$ represents the sparse reward when the task is considered successful. Giving a potential-based reward shaping function as follows:*

$$F = \gamma \cdot \Phi(s_{t+1}) - \Phi(s_t)$$

*Let another MDP $M' = \langle \mathcal{S}, \mathcal{A}, \mathcal{T}, R + F, \gamma, \rho_0 \rangle$, then every optimal policy in $M'$ will also be an optimal policy in $M$ and vice versa.*

The proof of Theorem 4.1 is provided in Appendix A.1. Theorem 4.1 indicates that $R^{PRM}$ does not distort the learning of the policy by introducing "deceptive" or "short-sighted" behaviors due to reward shaping. The optimal policy in $M'$ remains focused on maximizing the original task objective, i.e. maximize success rate. Based on Theorem 4.1, we have the following corollary:

**Corollary 4.2** (Convergence Guarantee). *The introduction of potential-based reward shaping will not affect the convergence of reinforcement learning algorithms.*

Since the PRM reward has a theoretical convergence guarantee, we proceed to demonstrate its efficiency in convergence, following the work of [41] (Detailed proof provided in Appendix A.2).

**Theorem 4.3** (Convergence Efficiency). *Consider two MDPs: $M = \langle \mathcal{S}, \mathcal{A}, \mathcal{T}, R + F, \gamma, \rho_0 \rangle$ with Q-function $Q$ and $M' = \langle \mathcal{S}, \mathcal{A}, \mathcal{T}, R, \gamma, \rho_0 \rangle$. If we initialize the Q-function in $M'$ as follows:*

$$Q'_0(s, a) = Q_0(s, a) + \Phi(s)$$

*Then for all timesteps $t \geq 0$ and $(s, a) \in \mathcal{S} \times \mathcal{A}$:*

$$Q'_t(s, a) = Q_t(s, a) + \Phi(s)$$

*where $Q_t$ and $Q'_t$ are the Q-functions of MDP $M$ and $M'$ respectively.*

The mathematical equivalence between PRM and $Q$-value initialization, as established in Theorem 4.3, provides two fundamental insights into why PRM accelerates convergence. First, it demonstrates that PRM effectively provides prior knowledge-based initialization for value functions, directly reducing the initial approximation error $|Q_0 - Q^*|$. Second, the proof formally justifies that any well-designed potential function $\Phi(s)$ serves as an actionable curriculum, where the shaping term $\gamma\Phi(s') - \Phi(s)$ progressively guides exploration toward high-value regions without altering optimal policies.

These theoretical results establish PRM not only as a heuristic reward design but also as a mathematically grounded approach to guide the policy with optimality and convergence.

### 4.3 Reinforcement Learning Augmentation

Giving an MDP $M = \langle \mathcal{S}, \mathcal{A}, \mathcal{T}, R, \gamma, \rho_0 \rangle$, our framework augment it through both high-level planning and low-level reward shaping.

At the high level, the policy cannot directly process the natural language form of the *plan_list*. To address this, we employ SimCSE[42], a sentence encoder $\mathbf{E}$, to embed the subtask prior. Specifically, the subtask prior is expressed as:

$$\mathcal{P} : prior = \mathbf{E}(plan\_list[\Psi(s)])$$

where subtask prior is the embedding of current subtask corresponding to state $s$. The state space is then augmented to $\mathcal{S} + \mathcal{P}$. At the low level, we use the reward $R^{PRM}$ from Section 4.2 to efficiently guide the learning of policy. Finally, the augmented MDP is expressed as:

$$\mathcal{M}' = \langle \mathcal{S} + \mathcal{P}, \mathcal{A}, \mathcal{T}, R^{PRM}, \gamma, \rho_0 \rangle$$

This formulation integrates both high-level planning through the subtask prior and low-level control via the PRM reward to guide policy learning. Consequently, we can utilize existing RL optimization algorithms to optimize the policy under the augmented MDP $M'$. In this paper, we use Soft Actor-Critic (SAC)[43] and Proximal Policy Optimization (PPO)[44].

## 5 Experiment

In this section, we evaluate PRM4RL on a diverse suite of long-term, sparse reward RL tasks, testing its ability to provide high-level plan and guide low-level training.

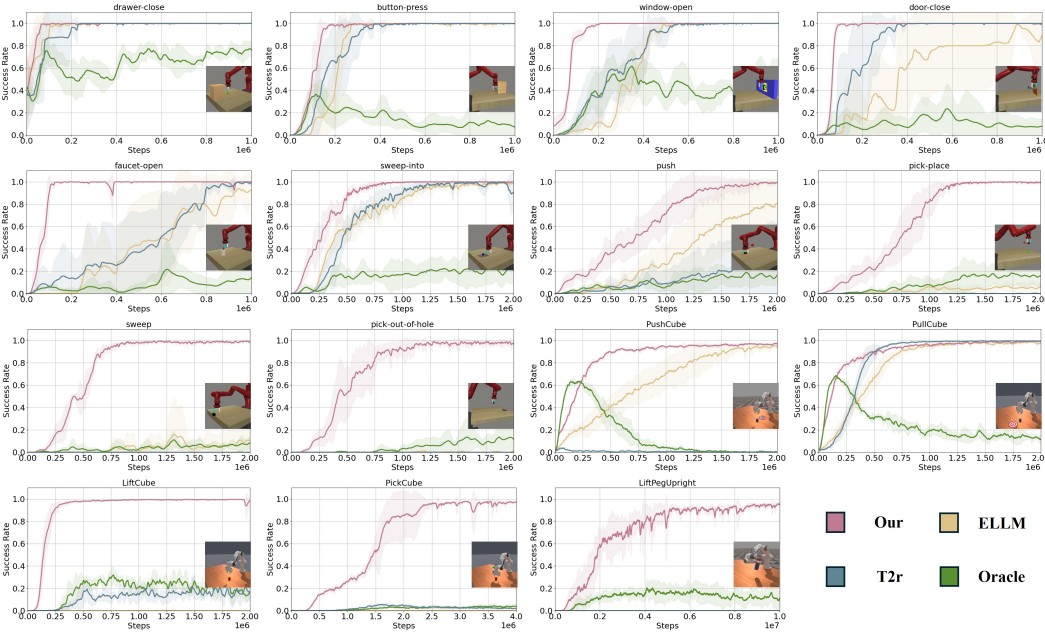

Figure 3: The learning curves of comparison methods on Metaworld and Maniskill measured by success rate. All experiments are conducted with 5 random seeds.

## 5.1 Experiment Settings

The framework is implemented based on the Stable-Baselines3 framework[45], ensuring consistency across all methods. We employ the same RL algorithm with identical training parameters for all methods. For all LLM-related requirements, we utilize OpenAI's GPT-4o[46]. Details of the hyper-parameters are provided in Appendix A.3. To ensure robustness and reliability, all experiments are conducted with five different random seeds.

**Baselines.** ELLM(ICML23) [23] utilizes an LLM as a high-level planner, embedding subtask priors into the policy. It serves as an example of an LLM-planner approach. T2R(ICLR24) [24] combines LLMs and human feedback to generate dense reward functions for policy training. We use it as an example of an LLM-rewarder approach. Oracle is the expert-written dense reward function provided by the environment. The details of the baseline settings are provided in Appendix A.2.2.

**Environments** MetaWorld[27] is an open-source simulated benchmark features a Sawyer robot interacting with a tabletop setup that includes drawer, window, ball, faucet, door and many objects. Maniskill[28] is an advanced robotics simulation platform designed for high-fidelity manipulation tasks. Further details about the environment and the tasks are provided in Appendix A.4.

## 5.2 Evaluation Analysis

### 5.2.1 Outperform previous LLM approaches

As shown in Figures 3 and 4, our method consistently outperforms the comparison methods across nearly all environments, demonstrating faster convergence and superior performance. We observe the following advantages: (1) In complex long-term tasks such as MW-pick-out-of-hole and MS-LiftPegUpright, the baseline methods achieve less than 15% success, while our method achieves nearly 100% success, highlighting our ability to effectively solve challenging tasks. (2) When compared to ELLM, our approach demonstrates a significantly improved convergence speed, which underscores the efficiency of the Progress Reward Model in providing dense, theoretically guaranteed rewards for the low-level control policy. This results in more stable and faster learning compared to traditional LLM-based planning approaches. (3) In comparison with T2R, the subtask decomposition in our framework plays a critical role in simplifying the coding process. By breaking down complex

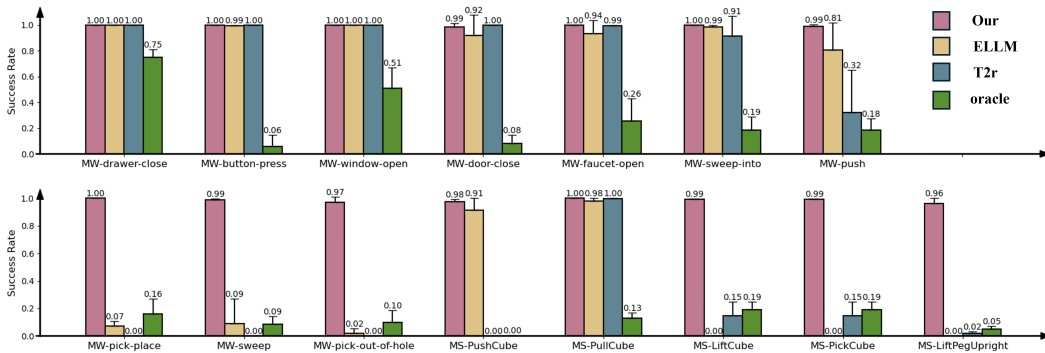

Figure 4: The evaluation results of comparison methods on Metaworld(With prefix 'MW') and Maniskill(with prefix 'MS') measured by success rate. All experiments are conducted with 5 random seeds.

tasks into more manageable subtasks, our method reduces the difficulty of understanding environment dynamics and designing useful metrics, enabling us to solve complex tasks that T2R fails to address. This advantage is particularly evident in tasks where the environment is more intricate and requires a clear separation of the planning and reward stages.

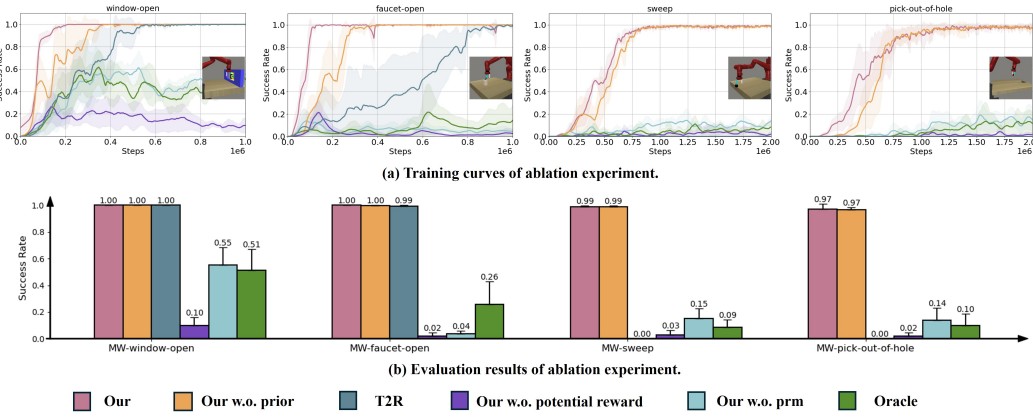

Figure 5: The evaluation results of ablation study on Metaworld measured by success rate. All experiment is conducted with 5 random seeds.

### 5.2.2 Ablation Study

To further validate the effectiveness of the two augmentations in our framework, we conduct an ablation study by constructing three additional baselines:

The results are shown in Figure 5. We observe the following findings:

(1) Compared with 'our w.o. prm', replacing the PRM reward with the oracle reward leads to a significant drop in performance. This demonstrates that, without an efficient and effective reward signal, directly feeding the subtask prior to the policy does not improve its performance. In most cases, the training curves and evaluation results of 'our w.o. prm' are similar to those of the oracle, with some instances where adding the subtask prior even causes a slight performance decrease.

(2) The performance of 'our w.o. prior' shows that abandoning the subtask prior results in slower convergence in the training curves. This highlights the role of the subtask prior in enabling efficient convergence for complex, long-term tasks. Moreover, 'our w.o. prior' still outperforms T2R, further demonstrating the effectiveness of the PRM reward in guiding the policy.

**w.o. PRM reward** (denoted as 'our w.o. prm'): The PRM reward is replaced with the oracle reward, while retaining the subtask prior.

**w.o. subtask prior** (denoted as 'our w.o. prior'): No subtask prior is provided, while retaining the PRM reward.

**w.o. potential-based reward shaping** (denoted as 'our w.o. potential reward'): The progress function is used directly as the reward, while retaining the subtask prior.

(3)The performance of 'our w.o. potential reward' further demonstrate the effectiveness of our potential-based reward model, where directly using progress as reward signal lead to a significant drop in performance. We also analyzed the anomalous behavior of 'our w.o. potential reward', where the success rate initially rises but then declines as training progresses. As shown in Figure 6, the episode reward increases, while the success rate paradoxically decreases. This suggests that the using progress function directly as reward function suffers from inefficiencies, as there are instances where the task fails despite receiving a high reward, leading the policy to learn in an undesirable direction. In contrast, our method ensures stable convergence, with the PRM providing well-structured and task-relevant rewards with optimality and convergence guarantees.

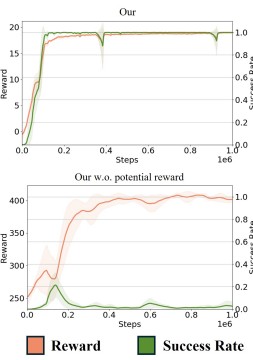

Figure 6: Training details of our and our w.o. potential reward.

In summary, the two enhancements in our framework—the subtask prior and the PRM reward—work synergistically to produce a "1+1 > 2" effect. Without the subtask prior, convergence speed is hindered, and without the reward signal, providing effective and efficient guidance becomes challenging.

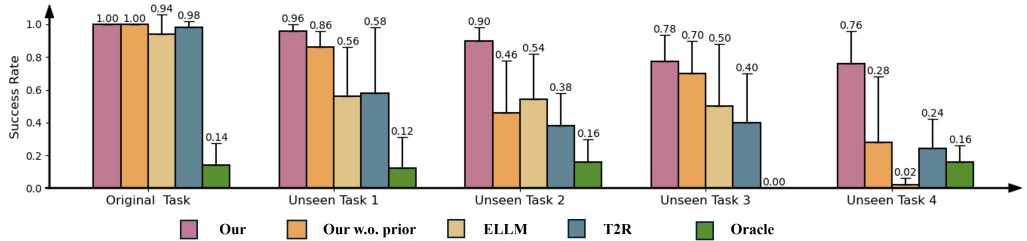

Figure 7: The evaluation results of generalization experiment. The results are averaged on 5 random seeds and 100 evaluations per seed.

### 5.2.3 Generalization Results

To evaluate the generalization capabilities of the trained policy, we test the trained policy's performance on unseen tasks in MetaWorld. Additional results and detailed information regarding the tasks are provided in Appendix A.2.1. As shown in Figure 7, while most methods perform well on the original task, we observe a performance decline on the unseen tasks. In comparison to other methods, our framework experiences the least performance loss, demonstrating relatively strong generalization abilities. Notably, when comparing to our 'w.o. prior' (our framework without the subtask prior), we observe a significant drop in performance on unseen tasks. This highlights the critical role of the subtask prior in enhancing generalization. The "verb + noun" pattern we designed effectively captures commonalities across different tasks, acting as a powerful prior that guides the policy to perform well on unseen tasks. Thus, the subtask prior serves as a robust tool for improving the generalization of reinforcement learning policies across a wide range of tasks.

## 6 Conclusion

In this paper, we introduced the Progress Reward Model for Reinforcement Learning (PRM4RL), a novel framework that integrates high-level planning and low-level reward shaping to address the challenges of long-term planning and sparse rewards in reinforcement learning. At the high level, we

decompose a complex task into manageable subtasks with subtask prior. At the low level, we design a Progress Reward Model with optimality and convergence guarantee. The experimental results demonstrate the effectiveness and efficiency of our method, significantly improving both convergence speed and the ability to solve complex, multi-stage tasks.

**Limitations.** Similar to T2R[24] and other methods[23, 21, 22] that depend on LLMs for planning or coding, our approach could be affected by LLM hallucinations during reasoning[47]. Besides, while our method demonstrates strong performance across various tasks, its applicability to domains outside of robotics remains unexplored. We anticipate that further optimization and adaptation of the framework will be necessary to address diverse environments and tasks.

## Acknowledgments and Disclosure of Funding

This work was supported by the National Natural Science Foundation of China (Grant No.62031001, Grant No.62325101).

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

# A Appendix

## A.1 Theoretical Proof

**Theorem A.1** (Policy Invariance). *Define an MDP $M = \langle \mathcal{S}, \mathcal{A}, \mathcal{T}, R, \gamma, \rho_0 \rangle$, where $R_t = \mathbb{I}(s_{t+1}) \cdot r_{bonus}$ represents the sparse reward when the task is considered successful. Giving a potential-based reward shaping function as follows:*

$$F = \gamma \cdot \Phi(s_{t+1}) - \Phi(s_t)$$

*Let another MDP $M' = \langle \mathcal{S}, \mathcal{A}, \mathcal{T}, R + F, \gamma, \rho_0 \rangle$, then every optimal policy in $M'$ will also be an optimal policy in $M$ and vice versa.*

*Proof.* For the original MDP $M$, we know that its optimal $Q$-function $Q_M^*$ satisfies the Bellman Equations

$$Q_M^*(s, a) = \mathbb{E}_{s' \sim P_{s,a}(\cdot)} \left[ R(s, a, s') + \gamma \max_{a' \in A} Q_M^*(s', a') \right]$$

Some simple algebraic manipulation then gives us

$$Q_M^*(s, a) - \Phi(s) = \mathbb{E}_{s'} \left[ R(s, a, s') + \gamma \Phi(s') - \Phi(s) + \gamma \max_{a' \in A} (Q_M^*(s', a') - \Phi(s')) \right]$$

If we now define $\hat{Q}_{M'}(s, a) \triangleq Q_M^*(s, a) - \Phi(s)$ and substitute that and $F(s, a, s') = \gamma \Phi(s') - \Phi(s)$ back into the previous equation, we get

$$\hat{Q}_{M'}(s, a) = \mathbb{E}_{s'} \left[ R(s, a, s') + F(s, a, s') + \gamma \max_{a' \in A} \hat{Q}_{M'}(s', a') \right]$$

$$= \mathbb{E}_{s'} \left[ R'(s, a, s') + \gamma \max_{a' \in A} \hat{Q}_{M'}(s', a') \right]$$

But this is exactly the Bellman equation for $M'$. For the undiscounted case, we moreover have $\hat{Q}_{M'}(s_0, a) = Q_M^*(s_0, a) - \Phi(s_0) = 0 - 0 = 0$. So, $\hat{Q}_{M'}(s, a)$ satisfies the Bellman equations for $M'$, and must in fact be the unique optimal $Q$-function. Thus, $Q_{M'}^*(s, a) = \hat{Q}_{M'}(s, a) = Q_M^*(s, a) - \Phi(s)$ and the optimal policy for $M'$ therefore satisfies.

$$*pi_{M'}^*(s) \in \arg\max_{a \in A} Q_{M'}^*(s, a) \tag{4}$$

$$= \arg\max_{a \in A} Q_M^*(s, a) - \Phi(s) \tag{5}$$

$$= \arg\max_{a \in A} Q_M^*(s, a) \tag{6}$$

* and is therefore also optimal in $M$. To show every optimal policy in $M$ is also optimal in $M'$, simply apply the same proof with the roles of $M$ and $M'$ interchanged (and using the shaping function $-F$). This completes the proof. $\square$

**Theorem A.2** (Convergence Efficiency). *Consider two MDPs: $M = \langle \mathcal{S}, \mathcal{A}, \mathcal{T}, R + F, \gamma, \rho_0 \rangle$ with $Q$-function $Q$ and $M' = \langle \mathcal{S}, \mathcal{A}, \mathcal{T}, R, \gamma, \rho_0 \rangle$. If we initialize the $Q$-function in $M'$ as follows:*

$$Q_0'(s, a) = Q_0(s, a) + \Phi(s)$$

*Then for all timesteps $t \geq 0$ and $(s, a) \in \mathcal{S} \times \mathcal{A}$:*

$$Q_t'(s, a) = Q_t(s, a) + \Phi(s) \tag{7}$$

*where $Q_t$ and $Q_t'$ are the $Q$-functions of MDP $M$ and $M'$ respectively.*

*Proof.* We prove the equivalence by induction on the update steps.

**Base case ($t = 0$):** By initialization definition:

$$Q'_0(s,a) = Q_0(s,a) + \Phi(s) \quad \forall(s,a) \tag{8}$$

**Inductive step:** Assume $Q'_k(s,a) = Q_k(s,a) + \Phi(s)$ holds for all $(s,a)$ at step $k$. Consider the Q-learning update for both agents when transitioning from $(s,a)$ to $(s',a')$ with reward $r$:

In MDP $M$:

$$Q_{k+1}(s,a) = Q_k(s,a) + \alpha_k\Big[r + F(s,s') + \gamma\max_{a'}Q_k(s',a') - Q_k(s,a)\Big]$$
$$= Q_k(s,a) + \alpha_k\Big[r + (\gamma\Phi(s') - \Phi(s)) + \gamma\max_{a'}Q_k(s',a') - Q_k(s,a)\Big]$$

In MDP $M'$:

$$Q'_{k+1}(s,a) = Q'_k(s,a) + \alpha_k\Big[r + \gamma\max_{a'}Q'_k(s',a') - Q'_k(s,a)\Big]$$
$$= [Q_k(s,a) + \Phi(s)] + \alpha_k\Big[r + \gamma\max_{a'}[Q_k(s',a') + \Phi(s')] - [Q_k(s,a) + \Phi(s)]\Big]$$
$$= Q_k(s,a) + \Phi(s) + \alpha_k\Big[r + \gamma\max_{a'}Q_k(s',a') + \gamma\Phi(s') - Q_k(s,a) - \Phi(s)\Big]$$

Simplify the difference:

$$Q'_{k+1}(s,a) - Q_{k+1}(s,a) = \Phi(s) + \alpha_k\Big[\gamma\Phi(s') - \Phi(s)\Big] - \alpha_k\Big[\gamma\Phi(s') - \Phi(s)\Big]$$
$$= \Phi(s)$$

Thus $Q'_{k+1}(s,a) = Q_{k+1}(s,a) + \Phi(s)$ maintains the invariant. By induction, the equivalence holds for all $t \geq 0$. $\qquad\square$

## A.2 Experiment Details

### A.2.1 Additional Generalization Results

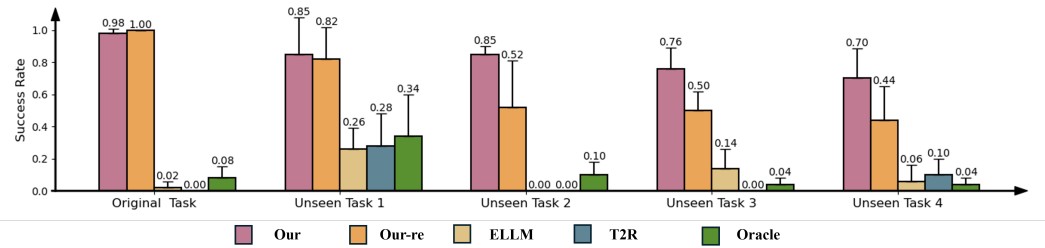

Figure 8: The evaluation results of generalization experiment.

We conduct additional generalization evaluations. Specifically, for the evaluation results shown in Section 5.2.3, the original task is 'Metaworld-faucet-open', the unseen tasks are 'Metaworld-drawer-close', 'Metaworld-button-press-wall', 'Metaworld-button-press', 'Metaworld-coffee-button', respectively. For the results shown in Figure A.2.1, the original task is 'Metaworld-pick-place', the unseen tasks are 'Metaworld-drawer-close', 'Metaworld-pick-place-wall', 'Metaworld-sweep-into', 'Metaworld-handle-press', respectively. We test the trained policy in evaluation tasks, and the results are averaged over 5 seeds and 100 test per seed.

### A.2.2 Baseline Settings

**ELLM**(ICML23)[23] utilizes an LLM as a high-level planner, embedding subtask priors into the policy. Since ELLM has not been previously tested on Metaworld and Maniskill, we re-implement it

the experiments. For goal embedding, we gave the same subtask prior as in our framework to the policy; For reward, we then gave it the subtask success reward, which is also goal-related as in its original application. We also use our $\Psi$ for determining current subtask to avoid enormous LLM calls.

**Text2Reward**(ICLR24)[24] utilizes an LLM to write dense reward function. As T2R has evaluate its performance on Metaworld and Maniskill, we directly reuse its generated reward functions. If there are few-shot and zero-shot versions of reward functions, we always choose the few-shot version for better performance. Specifically, for 'Metaworld-drawer-close', 'Metaworld-window-open', 'Metaworld-button-press', 'Metaworld-sweep-into', 'Metaworld-door-close', 'Maniskill-LiftCube', 'Maniskill-PickCube', we reuse its generated reward functions. For other tasks that are not orginally evaluated in T2R, we generate a reward function following its code using the same GPT-4o LLM.

### A.3 Hyper-Parameter Details

In this section, we provide the hyper-parameter details used for our reward function generation and reinforcement learning backbones. For progress function generation, we base on GPT-4o. In the experiments of the main body, the temperature of sampling is set to 0.7 for each experiment.

For reinforcement learning training, we employed the open-source PPO[44] and SAC implementation[43] from Stable-Baselines3[45] [2], and list the hyper-parameters in Table A.3 and A.3.

Table 1: Hyper-parameter of SAC algorithm applied to each task.

| Hyper-parameter | Value |
|---|---|
| Discount factor $\gamma$ | 0.99 |
| Target update frequency | 2 |
| Learning rate | $3 \times 10^{-4}$ |
| Train frequency | 1 |
| Soft update $\tau$ | $5 \times 10^{-3}$ |
| Gradient steps | 1 |
| Learning starts | 4000 |
| Batch Size | 512 |
| # of layers | 3 |
| Hidden units per layer | 256 |
| Initial temperature | 0.1 |
| Rollout steps per episode | 500 |

Table 2: Hyper-parameter of PPO algorithm applied to each task.

| Hyper-parameter | Value |
|---|---|
| Discount factor $\gamma$ | 0.8 |
| # of epochs per update | 15 |
| Learning rate | $1 \times 10^{-4}$ |
| # of environments | 8 |
| Batch size | 400 |
| Target KL divergence | 0.1 |
| # of layers | 3 |
| Hidden units per layer | 256 |
| # of steps per update | 3200 |

We utilize a 4 NVidia Geforce RTX-3090 graphic cards, 128 core CPUs, and 256 GiB memory server for RL training. The time required for training a policy is approximately 1 hours per task (5 seeds running simultaneouly) for MetaWorld, and between 1 to 10 hours for ManiSkill, varying with the task's difficulty.

---

[2]stable-baselines3 v2.6.0 (MIT License), code available at `https://github.com/DLR-RM/stable-baselines3`

## A.4 Tasks

In this section, we provide a full list of tasks within each simulation environment, accompanied by their corresponding language instructions. Across all tasks, we follow the default settings of their

### A.4.1 Metaworld

In METAWORLD[27][3] environment, we use a 7 DoF Sawyer robot arm with a fixed base to complete tabletop tasks. For all tasks, the observation space is a combination of the 3D position of the robot end-effector, a normalized measurement of gripper openness, the 3D position of the manipulated object, the quaternion of the object, all of the previous measurements, and the goal position. The environment adopts end-effector delta position control, which means the action space consists of the change of the end effector's 3D position, as well as the normalized torque the gripper should apply. For all tasks, the initial and target positions of the manipulated object and the initial joint positions of the robot arm are variable.

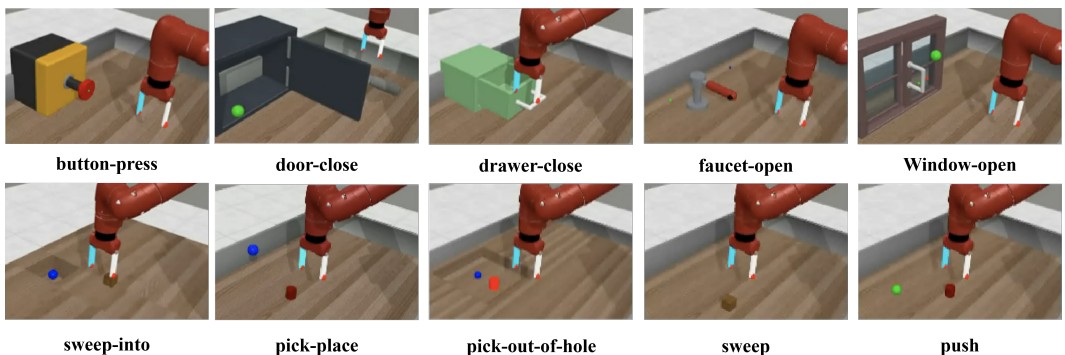

Figure 9: Tasks in Metaworld.

The task descriptions are provided in the following Table A.4.1.

Table 3: Task list of Metaworld

| Task | Task Description |
|---|---|
| button-press | Press a button in y coordination. |
| door-close | Close a door with a revolving joint by pushing the door's handle. |
| drawer-close | Close a drawer by its handle. |
| faucet-open | Rotate the handle counter-clockwise. |
| window-open | Push and open a sliding window by its handle. |
| sweep-into | Sweep a puck from the initial position into a hole. |
| pick-place | Pick up an object and move it to the goal location. |
| pick-out-of-hole | Pick an object out of a hole and move it to the goal location. |
| sweep | Sweep a puck off the table. |
| push | Push an object to the goal location. |

### A.4.2 Maniskill

MANISKILL[28][4] environment uses a 7 DoF Franka Panda as the default robot arm. For all tasks, the observation space consists of robot proprioception information (e.g. current joint positions, current joint velocities, robot base position and quaternion in the world frame) and task-specific information (e.g. goal position, endeffector position). We use end-effector delta pose control mode for this

---

[3]Metaworld v2.0.0 (MIT License), code available at `https://github.com/Farama-Foundation/Metaworld`

[4]Maniskill3 v3.0.0 (Apache-2.0 License), code available at `https://github.com/haosulab/ManiSkill`

environment, which controls the change of 3D position and rotation (represented as an axis-angle in the end-effector frame). For all tasks, the initial and target positions of the manipulated object, the initial joint positions of the robot arm and physical parameters (e.g. friction and damping coefficient) are variable.

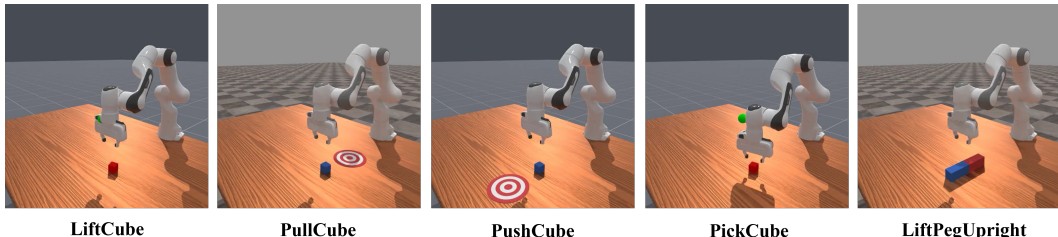

LiftCube    PullCube    PushCube    PickCube    LiftPegUpright

Figure 10: Tasks in Maniskill.

The task descriptions are provided in the following Table A.4.2.

Table 4: Task list of Maniskill

| Task | Task Description |
| --- | --- |
| LiftCube | Pick up cube A and lift it up by 0.2 meters. |
| PullCube | Pull a cube to the goal position. |
| PushCube | Push a cube to the goal position. |
| PickCube | Pick up cube A and move it to the 3D goal position. |
| LiftPegUpright | Move a peg laying on the table to any upright position on the table. |

## A.5  Prompt Details

To ground LLMs into robotics simulation environments, following Text2Reward[24] we propose a Pythonic prompt, which can be abstracted by the experts who developed the environment. This class-like prompt is a more compact representation than simply listing all environment attributes linearly, which can delete redundant information and save more tokens, and this Pythonic prompt can also better bootstrap Python code generation. More specifically, our prompt leverages python class, class attribute, python typing and comments to recursively define the environment. For simplicity, here we prompt the progress function to return the current subtask index together with the main progress.

Here we provide an example of the zero-shot prompt for Metaworld manipulation tasks:

**Basic Prompt** provides the foundational context for the task. The prompt establishes the user's role as an expert in robotics, reinforcement learning, and code generation, focusing on the development of a Python function called 'progress function' for RL. The objective is to help generate code that tracks progress during task execution, guiding the learning of a low-level policy.

> **Basic Propmt**
>
> You are an expert in robotics, reinforcement learning and code generation.We are going to use a robot arm to complete given tasks. Now I want you to help me write a python function named 'progress function' of reinforcement learning.

**Environment Information** defines the environment in which the robot operates. The BaseEnv class describes the robot and its interaction with objects in the environment, including their positions and states. The Robot and RigidObject classes further define the robot's end effector, object positions, and other relevant parameters. This context provides essential details for the task, enabling the LLM to make informed decisions about task progress and control.

```python
class BaseEnv(gym.env):
    self.robot : Robot # the robot in the environment
    self.obj : RigidObject # the first object in the environment
    self.goal_position : np.ndarray[(3,)] # indicate the 3D
        position of the goal
    near_object = float(tcp_to_obj <= 0.3)
    success = float(obj_to_target <= 0.07)

class Robot:
    self.ee_position : np.ndarray[(3,)] # indicate the 3D
        position of effector
    self.hand_init_pos: np.ndarray[(3,)] # indicate the initial
        3D position of effector
    self.tcp = self.ee_position - [0, 0, 0.045]  # indicate the
        3D position of effector's tool center point

class RigidObject:
    self.position : np.ndarray[(3,)] # indicate the 3D position
        of the rigid object
    self.quaternion : np.ndarray[(4,)] # indicate the quaternion
        of the rigid object
    self.obj_init_pos : np.ndarray[(3,)] # indicate the initial
        3D position of the rigid object
```

**Task Information** provides a description of the task to be completed by the robot. The task description, which needs to be filled in by the user, is crucial for decomposing the task into manageable subtasks.

Task Information

{ Fill in the task description here. } (e.g. In pick-place, the robot need to pick up the object and move it to the goal position.)

**Basic Instructions** provide guidelines for writing the code necessary for the task. The instructions emphasize the importance of using existing Python packages only when necessary, avoiding the invention of new variables or attributes, and incorporating step-by-step reasoning with clear comments. These instructions ensure that the code remains efficient, understandable, and aligned with the task's requirements.

Basic Instruction

1. You are allowed to use any existing python package if applicable. But only use these packages when it's really necessary.
2. Do not invent any variable or attribute that is not given.
3. Think step by step, add comment for reasoning and thought when you write code.

**Reasoning Instructions** outline a step-by-step procedure for task decomposition and the generation of key functions.

Reasoning Instructions

Follow the steps below:
a. Decompose the task and generate plan_list
a.1 Decompose the given task into a series of subtasks. Each subtask should be described in a 'verb + noun' form. tips: 'move to object', 'grasp object', 'move to goal'...
a.2 Generate a python list named plan_list where each subtask is an element of the plan_list.

b. Write a 'determination_function' for determine which subtask we are in based on Environment Information above.

b.1 For each subtask in the plan_list, consider when each subtask is considered completed, express the condition in python code form. Typical completion conditions: -'Move' completion: distance under threshold -'Grasp' completion: object's z-coordinate higher than its initialize z-coordinate.

b.2 Write a 'determination_function' using 'if' and 'elif' to categorize the current subtask based on Environment Information - consider the logic of determining subtasks completion, using 'if not' 'elif not' if needed.

b.3 The 'determination_function' return the current subtask's index in plan_list.

c. Write a 'progress_function'

c.1 For each subtask, construct a progress metric named 'subprogress' indicating the progress of the corresponding subtask. The 'subprogress' should be range from 0 (no progress) to 1 (completed) Typical subprogress metric: the distance between the robot's gripper and target object normalized by its initial distance;

c.2 Based on the 'determination_function', add the 'subprogress' computation codes in the corresponding 'if' or 'elif' block.

c.3 Add the 'main_progress' computation codes in each subtask's 'if' or 'elif' block. - 'main_progress' is the progress of the entire task. - typically, main_progress can be calculated by adding the current subtask idx and current subprogress, regarding all completed subprogress as '1'. The completed progress can be set to '1' without computing again.

c.4 'progress function' return the main_progress and the current subtask's index in plan_list.

## A.6 Example of Generated Progress Function

In this section, to help better understand the generation ability and results, we provide zero-shot generated reward function on Metaworld. The response of the prompt defined in Appendix A.5 is as follows:

LLM response in Metaworld-pick-place

```python
import numpy as np
# Plan the task into subtasks
def generate_plan_list():
    # Task decomposition: [verb + noun]
    plan_list = [
        'move to object',
        'grasp object',
        'move to goal',
        'release object'
    ]
    return plan_list

# Determining the current subtask based on environment
    information
def determination_function(env, plan_list):
    # Decompose the task completion into conditions for each
        subtask

    # Subtask 1: 'move to object'
    if np.linalg.norm(env.robot.tcp - env.obj.position) > 0.3:
        return 0  # Still moving towards the object

    # Subtask 2: 'grasp object'
    elif env.obj.position[2] <= env.obj.obj_init_pos[2]:
        return 1  # Object not yet grasped

    # Subtask 3: 'move to goal'
```

```python
        elif np.linalg.norm(env.obj.position - env.goal_position) >
            0.07:
            return 2  # Still moving towards the goal

    # Subtask 4: 'release object'
    else:
        return 3  # Task is completed

# Progress computation for each subtask
def progress_function(env):
    plan_list = generate_plan_list()
    subtask_idx = determination_function(env, plan_list)

    # Initialize variables
    main_progress = 0.0  # Overall task progress
    subprogress = 0.0  # Progress of the current subtask

    # Subtask 1: 'move to object'
    if subtask_idx == 0:
        # Progress of moving towards the object
        initial_distance = np.linalg.norm(env.robot.tcp - env.
            obj.obj_init_pos)
        current_distance = np.linalg.norm(env.robot.tcp - env.
            obj.position)
        subprogress = 1 - (current_distance / initial_distance)
        main_progress = subtask_idx + subprogress  # Task
            progress is the sum of subtasks

    # Subtask 2: 'grasp object'
    elif subtask_idx == 1:
        # Progress of grasping the object (object lifted above
            initial position)
        subprogress = (env.robot.ee_position[2] - env.obj.
            obj_init_pos[2]) / 0.5  # Normalize based on height
        main_progress = subtask_idx + subprogress

    # Subtask 3: 'move to goal'
    elif subtask_idx == 2:
        # Progress of moving towards the goal position
        initial_distance_to_goal = np.linalg.norm(env.obj.
            position - env.obj_init_pos)
        current_distance_to_goal = np.linalg.norm(env.obj.
            position - env.goal_position)
        subprogress = 1 - (current_distance_to_goal /
            initial_distance_to_goal)
        main_progress = subtask_idx + subprogress

    # Subtask 4: 'release object'
    elif subtask_idx == 3:
        # Task is complete, no further progress is needed
        subprogress = 1.0
        main_progress = 1.0

    return main_progress, subtask_idx
```

