# OpenReview forum: "Progress Reward Model for Reinforcement Learning via Large Language Models"
_NeurIPS.cc/2025/Conference — NeurIPS 2025 poster_

### Official Review · Reviewer_DqUw · 2025-06-08

**Clarity:** 3
**Significance:** 2
**Originality:** 2
**Rating:** 4
**Confidence:** 3

**Summary:**

**Summary**

This paper introduces **PRM4RL (Progress Reward Model for Reinforcement Learning)**, a novel framework designed to tackle the challenges of long-term, sparse-reward tasks in reinforcement learning. The core idea is to integrate the strengths of large language models (LLMs) in high-level planning with dense, low-level reward shaping to enable more efficient policy learning.

At the **high level**, PRM4RL decomposes complex tasks into subtasks using natural language priors and evaluates progress through a fine-grained progress function. At the **low level**, it constructs a dense reward signal inspired by potential-based reward shaping, leveraging the progress function as a potential function to provide theoretical guarantees on convergence and optimality.

The approach bridges the gap between previous LLM-based methods that focused either on planning or reward shaping alone. Extensive experiments on robotics control tasks (MetaWorld and ManiSkill) demonstrate that PRM4RL achieves state-of-the-art performance, improving task success rates and reducing reliance on frequent LLM calls.

Overall, PRM4RL provides a unified and theoretically grounded framework that enhances both planning and policy optimization in complex environments.

**Questions:**

### **Questions and Suggestions for the Authors**

1. **Clarification on Progress Function Construction**
   Could the authors provide more details on how the fine-grained progress functions are constructed across different tasks? Specifically, are they hand-crafted per task, or is there an automated or generalizable method for defining them? The generality of the method would impact its applicability to new domains.
   _Stronger evidence of automation or generalization would positively impact the evaluation score._

2. **Effect of LLM Errors on Performance**
   Given the reliance on LLMs for both task decomposition and prior knowledge extraction, how robust is the method to LLM hallucinations or errors in subtask generation? Have the authors evaluated how performance degrades when LLM output is noisy or incorrect?
   _A discussion or ablation showing robustness to LLM noise would strengthen the paper._

3. **Computational Efficiency and LLM Usage**
   While the method claims to reduce LLM queries, could the authors provide quantitative data on LLM calls and overall compute cost compared to baselines? How does the method scale with more complex tasks or larger LLMs?
   _A clearer analysis of compute/efficiency tradeoffs would make the contribution more practically compelling._

**Ethical Concerns:**

["NO or VERY MINOR ethics concerns only"]

**Final Justification:**

The author’s response addressed the majority of my questions and concerns; I am maintaining my original score.

**Limitations:**

Yes, the authors have explicitly acknowledged two key limitations:

LLM Hallucinations:

They note that, similar to prior work, their framework may be affected by hallucinations in large language models during reasoning and task decomposition.

Domain Generalization:
The current evaluation is limited to robotic control tasks, and the method's applicability to domains beyond robotics (e.g., NLP, games, real-world planning tasks) remains unexplored.

**Quality:**

3

**Strengths And Weaknesses:**

### **Quality**

**Strengths:**
- The paper addresses a well-known and practically significant challenge in reinforcement learning: sparse rewards in long-horizon tasks.
- The proposed framework, PRM4RL, integrates high-level task decomposition with low-level reward shaping using a theoretically grounded approach based on potential-based reward shaping.
- The empirical evaluation is thorough, covering multiple benchmarks (MetaWorld, ManiSkill) and demonstrating improved performance over both LLM-as-planner and LLM-as-rewarder baselines.

**Weaknesses:**
- It is unclear how sensitive the progress function construction is to task design or domain-specific heuristics. This could impact reproducibility and generalization.
- Some ablation studies are missing—for example, analyzing the effect of planning quality or progress estimation accuracy on final performance.

---

### **Clarity**

**Strengths:**
- The paper is generally well-written and easy to follow, especially in describing the motivation and contrast with prior work.
- Figures are informative and help visualize the proposed framework and comparisons.

---

### **Significance**

**Strengths:**
- PRM4RL proposes a unified framework that advances the integration of LLMs into RL in a more holistic manner than prior one-sided approaches.
- It contributes to an important research direction—LLM-augmented RL—by demonstrating how progress tracking can inform both planning and reward shaping.
- Potential impact is high in robotics and other real-world domains where sparse reward signals are common.

**Weaknesses:**
- Current experiments are limited to robotics. Broader applicability in domains like navigation, games, or instruction following remains unexplored.
- The reliance on pre-trained LLMs might limit deployment in resource-constrained settings.

---

### **Originality**

**Strengths:**
- The idea of using a fine-grained **progress function** to bridge high-level planning and low-level reward shaping is novel and compelling.
- The paper contributes a new hybrid paradigm to the LLM-RL literature, going beyond the typical planner/rewarder dichotomy.

**Weaknesses:**
- While the integration is novel, the individual components (e.g., reward shaping, LLM planning) draw heavily on existing literature.

---

> ### Author Rebuttal · Authors · 2025-07-31
>
> # Response to Reviewer DqUw
>
> We sincerely thank Reviewer DqUw for the thorough and supportive review. We are grateful for the recognition that our framework is **"novel and compelling,"** contributes a **"new hybrid paradigm"** to the literature, and provides a **"unified and theoretically grounded"** solution to a significant challenge in RL. We welcome this opportunity to provide the requested clarificationsto the insightful questions below.
>
> ---
> ## **Regarding Weaknesses**
> ### **1. Weakness from Quality Section**
> > **Reviewer's Comment 1:**
> > "It is unclear how sensitive the progress function construction is to task design or domain-specific heuristics."
>
> **Our Response:** We address the two aspects of sensitivity below:
> * **Robustness to Task Design:** Our framework's robustness to diverse task designs is empirically validated by its successful generation of high-quality functions for **all 15 diverse tasks** in our experiments. By combining the LLM's world knowledge with our structured environment description, our method can reliably generalize across a variety of tasks.
> * **Sensitivity to Domain-Specific Heuristics:** Our framework is designed to minimize sensitivity to ad-hoc heuristics by handling domain knowledge in a systematic way. We systematically encode the necessary domain knowledge into a **reusable, pythonic, class-based "API"** (e.g., `BaseEnv`, `Robot`) that code-specialized LLMs are adept at processing (see `Appendix E` ). This structured approach ensures that the generation process is reproducible and not dependent on fragile, task-specific prompt engineering.
>
> > **Reviewer's Comment 2:**
> > "Some ablation studies are missing—for example, analyzing the effect of planning quality or progress estimation accuracy on final performance."
>
> **Our Response:** This is a very insightful suggestion. While directly ablating the "quality" or "accuracy" of LLM-generated components is a challenging research problem, we believe a compelling proxy for the **"effect of planning quality"** can be found in our existing comparison against the Text2Reward (T2R) baseline.
> * Our PRM4RL operates on a **"high-quality" plan,** where a complex task is decomposed into a sequence of subtasks.
> * Text2Reward, by contrast, operates on a **"low-quality" plan,** by treating the task as a single, monolithic block.
>
> We highlight some results here:
> |Method|MW-button-press|MW-pick-place|MS-LiftPegUpright|
> |:-:|:-:|:-:|:-:|
> |Our Method|1.00|1.00|0.96|
> |Text2Reward|1.00|0.00|0.02|
>
> This significant performance gap on multi-stage, complex tasks(`pickplace`, `LiftPegUpright`) powerfully demonstrates the critical **impact of high-quality planning on final performance.** We agree with reviewer that a more detailed analysis of both planning quality and progress estimation accuracy is a valuable direction for future work.
> ### **2. Weakness from Significance Section**
> > **Reviewer's Comment 1:**
> > "Current experiments are limited to robotics. Broader applicability in domains like navigation, games, or instruction following remains unexplored."
>
> **Our Response:** We thank the reviewer for this insightful comment. We focus on robotics as it is an ideal testbed for the long-horizon, sparse-reward RL challenges our framework is designed to solve. **The core principle of our work, however, is domain-agnostic:** it applies to any task that can be logically decomposed into sub-goals whose progress can be measured from state variables.
>
> For instance, in games like `Minecraft`, our framework could auto-generate a dense reward signal to guide an agent through a complex crafting tree (e.g., find wood -> craft table -> craft pickaxe), a classic challenge in that domain. We believe these are promising future directions and will add a discussion of this broader potential to our paper.
>
> > **Reviewer's Comment 2:**
> > "The reliance on pre-trained LLMs might limit deployment in resource-constrained settings."
>
> **Our Response:**
>
> While the base LLM itself can be resource-intensive, our framework is extremely **resource-efficient** at training and deployment time.
> This is achieved through our **"One-Time Generation"** paradigm, where the LLM is invoked only once before training begins. During the training process, the agent is guided exclusively by the generated, lightweight Python function, **with no further LLM calls required.**
> Please refer to **`response to Question #3`** for a comprehensive, data-driven analysis of this efficiency.
> ### **3 Weakness from Originality Section**
> > **Reviewer's Comment:**
> > "While the integration is novel, the individual components (e.g., reward shaping, LLM planning) draw heavily on existing literature."
>
> **Our Response:**
>
> We are grateful for the **reviewers' recognition on our novelty and originality.** We completely agree that our primary novelty lies in the **synergistic integration** of theoretically-grounded concepts.
>
> Our contribution is the creation of what the reviewer aptly describes in the review as a **"new hybrid paradigm"**: an automated systemthat that connects high-level planning and low-level reward generation, distinguished by its theoretical guarantee. We are grateful for the recognition of this central contribution.
>
> ---
> ## **Regarding Questions**
> ### **Question 1: Clarification on Progress Function Construction**
> **Our Response:**
>
> Thank you for this excellent question, as it allows us to clarify a core novelty of our framework. To be clear, the progress functions Φ(s) are **not hand-crafted,** but are **fully and automatically generated by our LLM in a zero-shot manner.** This is achieved via our structured, Pythonic prompt (see `Appendix E`), where we define the environment with pythonic classes like `BaseEnv` and `Robot`, provide the LLM with a clear, semi-formal API of the world it needs to reason about.
>
> To generate a function for a new task, a user only needs to provide a high-level, natural language task description. The LLM then uses this structured context to write the complete, fine-grained progress function. A concrete example of a progress function Φ(s) generated with this **automated, zero-shot process** is provided in `Appendix F`. The successful **application across 15 diverse tasks** in our paper is the empirical validation of this generalizable method.
>
> ### **Question 2: Effect of LLM Errors on Performance**
> **Our Response:**
>
> Our primary approach to LLM hallucinations is not passive, reactive recovery, but **proactive mitigation** through a two-pronged strategy:
> 1. **Framework Structure:** By **decomposing a complex, long-horizon problem into a series of simple, manageable subtasks**, we fundamentally reduce the complexity of the code generation challenge. This is validated by our method's success on multi-stage tasks (e.g., 1.00 success on `MW-pick-place`) where the monolithic Text2Reward approach fails (0.00 success).
> 2. **Structured Prompting:** For each simple subtask, our **structured, Pythonic prompt** communicates the task in a format that code-specialized LLMs are best at understanding. By providing a clear, class-based "API" of the environment (see `Appendix E`), we **minimize ambiguity** and guide the LLM to generate code within this well-defined, familiar structure. **A new ablation experiments** confirms its importance: replacing it with a standard natural language prompt dropped the generation success rate to just 30%, demonstrating our approach is crucial for reliability.
>
> In practice, we found this two-pronged approach to be highly reliable, leading to the successful generation of valid, high-quality functions for **all 15 diverse tasks** in our experiments.
>
> As a **new ablation for robustness to LLM noise,** we increased the LLM's sampling temperature to 0.9. The generation success rate remained 100% and the resulting policy's performance was unaffected, confirming our framework's robustness.
>
> * **Limitations and Future Work:** We see great potential in **synergizing our method with iterative refinement systems** like Eureka [1]. PRM4RL could provide **a high-quality "perfect initialization"** for such systems, drastically reducing their search space. We believe this combination is a promising direction for future work.
>
>   References: [1] Ma, Yecheng Jason, et al. "Eureka: Human-level reward design via coding large language models." arXiv preprint arXiv:2310.12931 (2023).
> ### **Question 3: Computational Efficiency and LLM Usage**
> **Our Response:**
> 1. **Cost Analysis:**
>    Our method's cost is a single, upfront expense: the one-time call consumes 875 input tokens and 628 output tokens, with a wall-clock time of 9.776 seconds.
>
>     Our "One-Time Generation" is a fundamentally new and more efficient LLM Usage Pattern. The table below provides a clear cost-benefit analysis. Our method achieves the **best performance with the lowest cost**.
>
>     |Method|LLM Usage Pattern| Total LLM Calls(Typical 1M-step Training) |Average Success Rate on Metaworld/Maniskill |
>     |:-:|:-:|:-:|:-:|
>     |Our Method|One-Time|1|0.99 / 0.98|
>     |Text2Reward|Iterative, Human-in-the-Loop Refinement |N (N $\geqslant$ 1 cycles)|0.62 / 0.26|
>     |ELLM|Continuous, Per-Timestep Planning|$\approx 1000000$|0.68 / 0.38|
> 2. **Scalability Analysis:**
>    * **With Task Complexity:** Our hierarchical approach ensures that even for a complex task, the LLM only needs to generate progress metrics for simple, decomposed subtasks. The successful application of long-horizon tasks like `MS-LiftPegUpright` is empirical evidence of this scalability.
>    * **Scaling with LLM Size:** Our framework has a high performance ceiling (benefiting from larger models like GPT-4o) and a high floor. Our new experiments confirm its robustness to scaling down, showing that even a smaller, open-source 8B model (Qwen3-8B) can reliably produce high-performing policies (90% generation success rate; 0.99 task success rate).

---

### Official Review · Reviewer_nbNu · 2025-07-02

**Clarity:** 3
**Significance:** 2
**Originality:** 2
**Rating:** 4
**Confidence:** 4

**Summary:**

The paper proposes PRM4RL, a framework that combines high‑level task decomposition and low‑level dense reward shaping to tackle long‑horizon, sparse‑reward robotics tasks. An LLM first decomposes a complex task into a list of natural‑language subtasks, then auto‑generates a subtask determination function that maps raw states to the current subtask index, and a progress function whose temporal difference serves as a potential‑based shaping term. The resulting progress reward model provides dense rewards for the task. Empirical results on 10 MetaWorld and 5 ManiSkill manipulation tasks show faster convergence and higher success rates than prior LLM‑planner and LLM‑rewarder baselines, often matching or surpassing expert “oracle” rewards and requires fewer LLM calls.

**Questions:**

N/A

**Ethical Concerns:**

["NO or VERY MINOR ethics concerns only"]

**Final Justification:**

The authors addressed most of my concerns regarding novelty and comparison to previous work, so I increased my rating.

**Limitations:**

yes

**Quality:**

2

**Strengths And Weaknesses:**

Strengths
- The proposed method bridges planning and reward shaping. Unlike previous work that addresses only one side, the proposed method uses the LLM once to obtain both a high‑level plan and a dense reward, reducing engineering effort.
- Results on 15 challenging manipulation tasks are good. The proposed method has near‑100 % success on cases where previous methods have < 40 % performance. The proposed method also converges 2‑3× faster than baselines.
- The ablation is insightful. Removing either subtask priors or the PRM reward degrades performance.
- The learned policies generalize better to unseen tasks than baselines.

Weaknesses
- The proposed method seems like a combination of LLM-planner and Text2Reward.
- The scope is narrow. All experiments are simulated robotics with dense state access. The trained policy can only do one task. This is very different from recent vision‑language-actions models.
- The proposed method, similar to previous methods, relies heavily on environment knowledge.
- Text2Reward functions are reused from prior work, which is done two years ago with a much weaker language model.

---

> ### Author Rebuttal · Authors · 2025-07-26
>
> # Response to Reviewer nbNu
>
> We sincerely thank Reviewer nbNu for their concise and detailed review. We are especially grateful for the reviewer's thorough summary of our work's strengths and found our results on 15 challenging tasks to be **"good"** with **"near-100% success"** where prior methods fail, **"2-3x faster"** convergence, and **better generalization**. We also appreciate that they found our ablation study **"insightful"**.
>
> Given this strong positive assessment of our empirical contributions, we understand that the reviewer's primary concerns are the conceptual weaknesses raised. We welcome this opportunity to address each of these points directly.
>
> ---
>
> ### **Weakness 1**
> > **Reviewer's Comment:**
> > "The proposed method seems like a combination of LLM-planner and Text2Reward."
>
> **Our Response:**
>
> We appreciate the reviewer situating our work with respect to these two important paradigms. However, we respectfully argue that PRM4RL is not a simple combination, but rather a **novel, unified framework** that synergistically integrates planning and reward shaping. Our novelty is twofold:
>
> 1. **Theoretically-Grounded Reward:**
>    PRM4RL is the **first framework to introduce and formalize the use of potential-based reward shaping** within LLM-augmented RL. Unlike prior heuristic approaches, our method constructs a reward model with **proven convergence guarantees,** ensuring robust and optimal policy learning.
>
> 2. **A Synergistic and Efficient Structure LLM Usage Pattern:**
>
>    We introduce a new, more efficient, and effective LLM usage pattern—**our "One-Time Generation" structure**—which requires only a single LLM call to generate the entire hierarchical setup. The following table contrasts this synergistic pattern with prior paradigms, highlighting the immense benefits in both efficiency and final performance:
>
>     | Method      | LLM Usage Pattern  | Total LLM Calls(Typical 1M-step Training) | Average Success Rate on Metaworld / Maniskill |
>     |:-:|:-:|:-:|:-:|
>     | Our Method  | One-Time | 1 | 0.99 / 0.98 |
>     | Text2Reward | Iterative, Human-in-the-Loop Refinement | N (N $\geqslant$ 1 cycles) | 0.62 / 0.26  |
>     | ELLM(LLM Planner) | Continuous, Per-Timestep Planning | $\approx 1000000$  | 0.68 / 0.38  |
>
> It is this combination of **theoretical guarantees and a novel, synergistic structure**—validated by our ablation studies —that makes our framework fundamentally more than a simple combination and represents our core contribution.
>
> ### **Weakness 2**
> > **Reviewer's Comment:**
> > "The scope is narrow. All experiments are simulated robotics with dense state access. The trained policy can only do one task. This is very different from recent vision language-actions models."
>
> **Our Response:**
>
> We thank the reviewer for raising this important point, as it allows us to clarify the specific research paradigm our work addresses and how it differs from the large-scale Vision-Language-Action (VLA) models.
>
> 1. **Clarifying the Research Paradigm:** Our work addresses the classic but critical challenge of **how to make Reinforcement Learning agents learn efficiently** in complex, long-horizon tasks with sparse rewards. Therefore, we followed standard practice in the RL community by using **a simple policy architecture (a 3-layer MLP)** and focused on its ability to master a single difficult task at a time. This is fundamentally different from the VLA paradigm, which focuses on achieving broad generalization through **large-scale imitation learning with complex architectures, massive parameter counts, and large-scale training.** The table below highlights this architectural difference:
>
>    | Method                         | Main Policy Structure | Hidden Dim | Policy Params       |
>    |:------------------------------:|:---------------------:|:----------:|:-------------------:|
>    | Our Method                     |  3-Layers MLP         | 256        | $\approx 0.0002$ B  |
>    | VLA (e,g., $\pi_0$ [1])        | 18-Layers Transformer | 2048       | $\approx 3.3$ B     |
>
>    This vast difference in scale underscores that our work and VLAs are designed to solve problems from different domain.
>
> 2. **On Generalization to Multiple Tasks:** We would like to clarify the point that "the trained policy can only do one task". We highlight our **zero-shot generalization results on unseen tasks** from Section 5.2.3, Figure 7 and Appendix B.1, Figure 8.
>
>    | Method|faucet-open(original)|drawer-close|button-press|coffee|
>    |:-:|:-:|:-:|:-:|:-:|
>    | Our Method     | 1.00| 0.96 | 0.90| 0.78 |
>    | Our w.o. prior | 1.00| 0.86 | 0.46| 0.70 |
>    |     ELLM       | 0.94| 0.56 | 0.54| 0.50 |
>    | Text2Reward    | 0.98| 0.58 | 0.38| 0.40 |
>
>     These results show that under the same 3-layer MLP policy structure, our trained policies demonstrate **significantly better generalization capabilities** compared to all baselines. This success is largely attributed to our **"verb + noun" subtask prior**, which enables the policy to transfer learned skills.
>
> 3. **A Synergistic Future with VLA Models:** Finally, we see these two research paradigms as complementary, not competitive. Our framework, which excels at **generating high-quality rewards for complex tasks,** could be invaluable for **efficiently fine-tuning a generalist VLA model** for a specific, novel task where expert data is scarce, bridging the gap between broad knowledge and specialized skills.
>
> We will expand on these clarifications in the final manuscript to better frame our work's contributions.
>
> Reference: [1] Black, Kevin, et al. "$\pi_0 $: A Vision-Language-Action Flow Model for General Robot Control." arXiv preprint arXiv:2410.24164 (2024).
>
> ### **Weakness 3**
> > **Reviewer's Comment:**
> > "The proposed method, similar to previous methods, relies heavily on environment knowledge."
>
> **Our Response:**
>
> The reviewer is correct that our method requires environment knowledge, which is true for most current approaches that apply LLMs to robotics. We view this not as a mere reliance, but as a necessary **"grounding"** of the LLM's abstract world knowledge into the specific physics and embodiment of the agent.
>
> Our framework is designed to make this grounding process as systematic and efficient as possible. Specifically:
>
> * **Structured Grounding:** As detailed in `Appendix E`, we achieve this grounding through a **structured, class-based Pythonic prompt.** By defining the environment with classes like `BaseEnv` and `Robot`, we provide the LLM with a clear, semi-formal "API" of the world it needs to reason about.
> * **One-Time Effort:** A key advantage of our framework is that this grounding process happens **only once** when designing the initial prompt. This is far more efficient than methods that might require continuous re-grounding or prompting during the learning process.
>
> Thus, while environment knowledge is necessary, our framework structures this knowledge in a reusable way and minimizes the interaction cost to a single upfront step.
>
> ### **Weakness 4**
> > **Reviewer's Comment:**
> > "Text2Reward functions are reused from prior work, which is done two years ago with a much weaker language model."
>
> **Our Response:**
>
> This is a very fair and insightful point regarding the importance of a strong baseline. We thank the reviewer for raising this, as it highlights an area where our manuscript was not sufficiently clear.
>
> We sincerely apologize for the ambiguity in our experimental description (`Appendix B.2`). We would like to clarify this important detail: for all tasks not originally evaluated in the Text2Reward (T2R) paper, we followed their methodology but **used the same state-of-the-art LLM, GPT-4o, that our own method uses.** Our intention was to create the strongest possible baseline and ensure a direct, 'apples-to-apples' comparison between the frameworks themselves, controlled for LLM capability.
>
> To make this comparison concrete, we highlight a few representative results from our experiments:
>
> | **Task**                 | **Our Method** | **Text2Reward With GPT-4o** |
> |:------------------------:|:--------------:|:---------------------------:|
> | Maniskill-PullCube       | 1.00           | 1.00                        |
> | Metaworld-push           | 0.99           | 0.32                        |
> | Maniskill-LiftPegUpright | 0.96           | 0.02                        |
> | Metaworld-pick-place     | 1.00           | 0.00                        |
>
> These results reveal a clear pattern. While the T2R framework can succeed on simpler, single-stage tasks (e.g., `MS-PullCube`), it struggles significantly on complex, multi-stage tasks that require hierarchical reasoning (e.g., `MW-pick-place`, `MS-LiftPegUpright`). Attempting to generate a single, monolithic reward function for such tasks is far more prone to failure, even for a powerful LLM like GPT-4o.
>
> The fact that PRM4RL still significantly outperforms this GPT-4o-powered T2R baseline—especially on complex, multi-stage tasks where it fails completely—therefore strongly supports our central claim: the **architectural advantages** of our unified, hierarchical framework are the key driver of its success, not simply the power of the underlying LLM.
>
> We are grateful to the reviewer for pushing us to make this crucial detail more transparent. We will revise the experimental section in the camera-ready version to state this explicitly and remove any ambiguity.

---

> > ### Comment · Reviewer_nbNu · 2025-08-03
> >
> > Thank you for your reply! It addressed most of my concerns on novelty and comparison to prior work, so I have increased my score to 4 (which will be visible to you later).

---

> > > ### Author Response · Authors · 2025-08-04
> > > **Thank You for Your Feedback and Engagement!**
> > >
> > > We sincerely thank the reviewer for their valuable feedback and for engaging with our rebuttal. Your critiques on novelty and scope were particularly helpful and have allowed us to significantly refine the paper's positioning. We are very grateful that this constructive dialogue led to your decision to raise your score.

---

### Official Review · Reviewer_ccSg · 2025-07-03

**Clarity:** 3
**Significance:** 3
**Originality:** 3
**Rating:** 5
**Confidence:** 3

**Summary:**

This paper presents the Progress Reward Model for Reinforcement Learning (PRM4RL), a novel framework that combines high-level task decomposition with low-level dense reward shaping to enhance RL learning efficiency in sparse reward tasks. The approach leverages LLMs to automatically generate subtask priors, a progress function for each subtask, and a determination function that tracks the agent’s position within a hierarchical task structure. With potential-based reward shaping, the method provides dense, informative feedback while preserving policy optimality and ensuring convergence by theoretical analysis. Extensive experiments on complex robotic manipulation tasks MetaWorld and Maniskill demonstrate that PRM4RL consistently outperforms previous LLM-based pure planning or pure reward-generation approaches in both learning performance and sample efficiency.

**Questions:**

See weakness 1 and 2.

**Ethical Concerns:**

["NO or VERY MINOR ethics concerns only"]

**Final Justification:**

I think the author's response solved most of my problems. Although the high-level ideas of this article belong to the same family as Eureka and text2reward and other series of works, I think the methods and experiments of this article are still decent methods with practical significance. I hope the author can add the concerns mentioned by all reviewers to the limitation or related discussions. I would like to raise my rating to 5.

**Limitations:**

yes

**Paper Formatting Concerns:**

no problem

**Quality:**

3

**Strengths And Weaknesses:**

Strengths:
1. This paper effectively combines the capabilities of LLMs as both a planner and a reward shaper, providing fine-grained task decomposition and reward guidance. It proposes an efficient method to minimize LLM calls by extracting both the task plan and logical functions within a single code-format prompt, instead of heavily per-step realtime LLM query.
2. The theoretical analysis demonstrated for optimal policy preservation and convergence with the proposed potential-based reward shaping, proving that the designed rewards will not lead to policy degradation or instability.
3. Extensive experiments show clear improvement over baselines, including ELLM (pure planner) and T2R (pure rewarder), particularly in sample efficiency and in solving long-horizon, sparse reward tasks where baselines often fail. The ablation study is well-justified to show the benefits of designed modules.

Weakness:
1. Although the framework is more efficient in LLM call frequency, it remains unclear if the LLM prompt design is robust across highly diverse environments or tasks and whether significant prompt engineering is needed for each new domain. It is not clear whether this fewer query method sacrifices the robustness of intensive real-time call methods to rare situations and tasks not seen during training.
2. In the ablation experiment in Fig. 5, there is almost no difference in the impact of removing the prior on performance, and it only slightly improves the training efficiency. Does it mean that the benefit of priors has been somehow encoded into the PRM, so the prior does not actually have a significant effect then. Conversely, this also indicates that the effect of PRM depends heavily on the design of the prior decomposition subtask. Can ablation experiments be added to better explain the relationship between prior and PRM?
3. The current formulation of PRM and plan lists rely on structured, geometry-driven metrics (e.g., position, distance related tasks). I'm interested in whether the progress function can generalize to tasks requiring complex reasoning or abstract metrics, such as improving movement fluency or something else. Can PRM be extended to evaluate progress in such scenarios?

---

> ### Author Rebuttal · Authors · 2025-07-26
>
> # Response to Reviewer ccSG
>
> We sincerely thank Reviewer ccSg for their thorough and positive review. We are particularly grateful for their clear and insigtful recognition of our framework's key strengths, including **the effective and efficient integration of planning and reward shaping**, **the strong theoretical guarantees**, and **the well-justified experimental results**. The reviewer's thoughtful questions regarding robustness, the role of the prior, and generalization to abstract metrics are highly insightful, and we welcome this opportunity to elaborate on these points and discuss the future potential of our framework.
>
> ---
>
> ### **Weakness 1**
> > **Reviewer's Comment:**
> > "Although the framework is more efficient in LLM call frequency, it remains unclear if the LLM prompt design is robust across highly diverse environments or tasks and whether significant prompt engineering is needed for each new domain. It is not clear whether this fewer query method sacrifices the robustness of intensive real-time call methods to rare situations and tasks not seen during training."
>
> **Our Response:**
>
> We thank the reviewer for these crucial questions about the practicality and robustness of our framework. We address each of the three points below.
>
> 1. **On Prompt Robustness for Diverse Tasks:** Our prompt is designed to be highly robust within a given domain by using a structured, class-based representation of the environment. As detailed in `Appendix E`, we define the environment's components using Python classes like `BaseEnv` and `Robot`. This structured format is highly effective as it communicates the task in a language that code-specialized LLMs excel at understanding. This ensures that for any new manipulation task, the **core prompt structure remains unchanged;** often, only the **one-line natural language task description needs to be updated.** The successful application of this prompt structure **across all 15 diverse tasks** is strong evidence of its robustness.
>
> 2. **On Prompt Engineering for New Domains:**
> The reviewer's point on prompt engineering is insightful. Our class-based design clarifies what this "engineering" entails.
> To adapt our framework to a completely new domain (e.g., from robotics to navigation), an expert would need to define a new set of classes that describe that domain's state and action spaces (see `Appendix E`). We argue that this is a **systematic, one-time effort to create a domain-specific API**, rather than the iterative work often associated with "prompt engineering." Our work provides a clear template for this robust approach.
>
> 3. **On the Efficiency vs. Robustness Trade-off:**
>    Our framework is designed to strike a highly practical balance that delivers both state-of-the-art robustness and feasible efficiency. The reviewer correctly identifies "intensive real-time call methods" as a theoretical alternative, but we argue this paradigm is **computationally prohibitive for standard RL training.** A typical training run of millions of steps would require an equal number of LLM calls, incurring infeasible latency and financial cost. The table below provides a clear cost-benefit analysis against the baselines from our paper:
>
>    | Method      | LLM Usage Pattern | Total LLM Calls(Typical 1M-step Training) | Average Success Rate on Metaworld / Maniskill |
>    |:-:|:-:|:-:|:-:|
>    | Our Method  | One-Time  | 1  | 0.99 / 0.98         |
>    | Text2Reward | Iterative, Human-in-the-Loop Refinement | N (N $\geqslant$ 1 cycles)   | 0.62 / 0.26         |
>    | ELLM        | Continuous, Per-Timestep Planning       | $\approx 1000000$ | 0.68 / 0.38         |
>
>    As the table shows, our approach is not just orders of magnitude more efficient, but it also delivers superior final performance.
>
>    Furthermore, regarding **robustness on "tasks not seen during training,"** our generalization experiments (`Section 5.2.3`, line 280~290, Figure 7) show that our framework's structured priors enable significantly better zero-shot performance on unseen tasks than baselines.
>
>    Finally, to **ensure robustness and mitigate LLM hallucination,** we employ a **proactive, two-pronged mitigation strategy:**
>    * **Framework Structure:** Our framework decomposes complex problems into simple subtasks, fundamentally reducing the generation difficulty. This is **empirically validated** by our method's success on multi-stage tasks (e.g., 1.00 success on MW-pick-place) where monolithic approaches like Text2Reward fail (0.00 success).
>    * **Structured Prompting:** Our structured, class-based prompt provides a clear "API" for the LLM, which is crucial for reliability (Appendix E). **A new ablation** confirms this: replacing our structured prompt with a standard natural language prompt caused the generation success rate to plummet to just 30%.
> ### **Weakness 2**
> > **Reviewer's Comment:**
> > "In the ablation experiment in Fig. 5, there is almost no difference in the impact of removing the prior on performance, and it only slightly improves the training efficiency. Does it mean that the benefit of priors has been somehow encoded into the PRM, so the prior does not actually have a significant effect then. Conversely, this also indicates that the effect of PRM depends heavily on the design of the prior decomposition subtask. Can ablation experiments be added to better explain the relationship between prior and PRM?"
>
> **Our Response:**
>
> We thank the reviewer for this close reading of our ablation study and for asking for a deeper explanation of the prior's role. Our existing experiments demonstrate its key contributions as follows:
>
>
> * **Impact on Convergence Speed:** As the reviewer noted, the prior improves training efficiency. The training curves in Figure 5a show that the policy without the subtask prior (`Our w.o. prior`) exhibits **slower convergence.** The prior provides essential high-level context that guides the policy search more efficiently.
>
> * **Critical Role in Generalization:** The prior's most critical contribution is revealed in our generalization experiments(Section 5.2.3, line 280~290, Figure 7). The results are summarized as follows.
>
>   | Method         | Original Task | Unseen Task 1 | Unseen Task 2 | Unseen Task 3 | Unseen Task 4 |
>   |:--------------:|:-------------:|:-------------:|:-------------:|:-------------:|:-------------:|
>   | Our Method     | 1.00          | 0.96          | 0.90          | 0.78          | 0.76          |
>   | Our w.o. prior | 1.00          | 0.86          | 0.46          | 0.70          | 0.28          |
>
>   We must first sincerely apologize for a confusing caption in the original Figure 7, where the label `Our-re` was used for our model without the prior (`Our w.o. prior`). When correctly interpreted, the results are unambiguous: the `Our w.o. prior` model shows **a significant drop in performance on all unseen tasks.** This demonstrates that the prior, encoded in our "verb + noun" format, is crucial for the policy to generalize its learned skills.
>
> **Synergy between Prior and PRM:** These results reveal that the prior and the PRM are synergistic, not redundant. The `PRM` provides the dense, low-level reward signal (what to do now), while the `prior` embedding provides the policy with high-level contextual information (what is the current abstract goal) . It is this context that enables both faster learning and, most importantly, better generalization.
>
> We believe our existing experiments already demonstrate this crucial relationship. We will correct the caption in Figure 7 and revise the text in Section 5.2.2 to articulate this synergy more explicitly in the camera-ready version
> ### **Weakness 3**
> > **Reviewer's Comment:**
> > "The current formulation of PRM and plan lists rely on structured, geometry-driven metrics (e.g., position, distance related tasks). I'm interested in whether the progress function can generalize to tasks requiring complex reasoning or abstract metrics, such as improving movement fluency or something else. Can PRM be extended to evaluate progress in such scenarios?"
>
> **Our Response:**
>
> This is an excellent and truly forward-thinking question. We are genuinely grateful to the reviewer for highlighting this future direction, which pushes beyond the standard geometric goals and explores the broader potential of our framework.
>
> The reviewer is correct that our current experiments focus on tasks where progress is intuitively measured by geometric metrics. However, the PRM4RL framework is **not inherently limited to these metrics.** The core mechanism is prompting an LLM to write a Python function for Φ(s) based on the provided state space description. If the state space included more abstract features related to movement quality, our framework is well-equipped to generate a corresponding progress metric.
>
> For example, to address the reviewer's insightful suggestion of **improving movement fluency,** we could include the robot's joint jerk (the time derivative of acceleration) in the state observation. Then, we could add a natural language instruction like: "Minimize the integral of the squared jerk during the movement." in the prompt. The LLM could then generate a subprogress function that take movement fluency into consideration.
>
> The reviewer's suggestion has inspired us to recognize that the true strength of our framework lies in its ability to **translate any high-level, language-described goal**—be it geometric, physical, or abstract like 'fluency'—**into a computable reward signal.** Exploring these more nuanced objectives is a key avenue for future research, for which our framework provides a strong and flexible foundation.
>
> We again thank the reviewer for this insightful and forward-thinking question. It has sparked what we feel is a valuable discussion, and **we would be very happy to engage in further dialogue on this topic during the upcoming discussion period.**

---

> > ### Comment · Reviewer_ccSg · 2025-08-01
> >
> > I think the author's response solved most of my problems. Although the high-level ideas of this article belong to the same family as Eureka and text2reward and other series of works, I think the methods and experiments of this article are still decent methods with practical significance. I hope the author can add the concerns mentioned by all reviewers to the limitation or related discussions.

---

> > > ### Author Response · Authors · 2025-08-04
> > > **Thank You for Your Feedback and Engagement!**
> > >
> > > We sincerely thank the **Reviewer ccSg**  for their time and for engaging with our rebuttal. We are very grateful for their positive assessment of our work's methods and practical significance.
> > >
> > > We will absolutely follow their excellent suggestion and will incorporate a thorough discussion of the concerns raised by all reviewers into the Limitations and Related Works sections of our final manuscript.
> > >
> > > Thank you again for your constructive and valuable feedback!

---

### Official Review · Reviewer_MTbw · 2025-07-05

**Clarity:** 3
**Significance:** 2
**Originality:** 2
**Rating:** 4
**Confidence:** 3

**Summary:**

The paper introduces PRM4RL, a framework that first asks an LLM to break a long-horizon robotics task into natural-language subtasks, then derives a per-subtask progress function Φ and turns its temporal difference into a potential-based shaping reward. The authors prove policy-invariance and convergence, embed the current sub-goal description into the state, and train SAC/PPO policies on Meta-World and ManiSkill. Experiments show higher success rates and fewer LLM calls than SayCan-style planners or Text2Reward-style rewarders.

**Questions:**

Besides the points of Weaknesses:
- Please compare to a hand-coded distance-to-goal or potential-based reward to isolate LLM contributions.
- What is the actual token and wall-clock cost per episode for PRM4RL? How does performance scale to 7B-parameter models?
- Does the same Φ generalize if the task dynamics change (e.g., object mass or friction)?
- How does the method detect or recover from a misleading or suboptimal Φ generated by the LLM?

**Ethical Concerns:**

["NO or VERY MINOR ethics concerns only"]

**Final Justification:**

I appreciate authors for their further feedback. I would be happy to increase my score and encourage authors to put the new results in the revisions.

**Limitations:**

The paper included limitations in the conclusion section.

**Paper Formatting Concerns:**

No.

**Quality:**

3

**Strengths And Weaknesses:**

**Strengths:**
  - Blends LLM-based planning and reward shaping in a way that’s conceptually sound and empirically useful.
  - Implementation is practical—program-aided prompting keeps LLM outputs structured and reproducible.
  - Outperforms published LLM-augmented RL methods on multiple simulated manipulation tasks, with clear ablation.

**Weaknesses:**
  - Potential-based shaping for hierarchical RL is well studied ([1], [4]), and recent LLM-based curricula like CurricuLLM ([3]) and ICIRA-24 skill shaping ([2]) tackle closely related problems but are not cited.
  - Only simulated robotic arms with proprioceptive state; no vision, no physical robots, and limited ablations (e.g., no hand-crafted potential or naive reward as control). Meanwhile, the submission does not provide codes to reproduce the experiments.
  - No analysis of how robust Φ is to environment changes.

**References**
- [1] “Policy Invariance under Reward Transformations.” ICML 1999.
- [2] “Utilizing Large Language Models for Robot Skill Reward Shaping in RL.” ICIRA 2024.
- [3] “CurricuLLM: Automatic Task Curricula Design for Learning Complex Robot Skills using Large Language Models.” ICRA 2025.
- 4] “Potential-Based Reward Shaping for Hierarchical Reinforcement Learning.” IJCAI 2015.

---

> ### Author Rebuttal · Authors · 2025-07-25
>
> # Response to Reviewer MTbw
>
> We thank Reviewer MTbw for their thoughtful review. We are grateful for their recognition that our framework is **"conceptually sound and empirically useful".** In response to their valid concerns, especially regarding the evaluation, we have conducted **new experiments** and provide detailed clarifications below. We hope this new evidence will resolve the main issues raised and demonstrate the significance of our contributions.
>
> Due to the character limit, we have kept our responses as concise as possible, but we warmly welcome the opportunity to elaborate on any of these points in greater detail during the discussion period.
>
> ---
>
> ### **Weakness 1**
> > **Reviewer's Comment:**
> > "Potential-based shaping for hierarchical RL is well studied ([1], [4]), and recent LLM-based curricula like CurricuLLM ([3]) and ICIRA-24 skill shaping ([2]) tackle closely related problems but are not cited."
>
> **Our Response:**
>
> Thank you for highlighting these important connections and references.
>
> * **On Our Novelty:** We agree that our work builds upon the strong foundation of potential-based reward shaping. Our novelty is twofold:
>   1. We are **the first framework to introduce and formalize** potential-based reward shaping within LLM-augmented RL, providing proven **convergence guarantees**.
>   2. We introduce a **highly efficient structure** that uses a single LLM call to synergistically generate the entire hierarchical setup: the plan, the potential function (Φ(s)), and a novel, zero-cost subtask tracker (Ψ(s)).
>
>   Therefore, while the underlying theory is classic, our contribution is its novel application and the creation of an efficient, automated framework. The significant performance of this principled integration is validated by our ablation studies.
>
> * **On Citations:** We appreciate the references and will add a discussion of of these relevant works to our Related Works section to better contrast our approach.
>
> ### **Weakness 2**
> > Reviewer's Comment:
> > "Only simulated robotic arms with proprioceptive state; no vision, no physical robots, and limited ablations (e.g., no hand-crafted potential or naive reward as control). Meanwhile, the submission does not provide codes to reproduce the experiments."
>
> **Our Response:**
>
> We thank the reviewer for these detailed points.
>
> 1. **On Code Availability:** We would like to respectfully clarify that our **full codebase was provided in the supplementary material**. We apologize if this was not sufficiently clear and will state it more prominently in the final version.
>
> 2. **On Ablations and Baselines:** We have conducted new experiments with additional hand-coded baselines. For the sake of brevity, we present new results and analysis in our response to **`Question #1`**.
>
> 3. **On Experimental Scope:**
> We would like to clarify that our framework is indeed **compatible with visual inputs**. In modern robotics, proprioceptive state often coexists with vision as "privileged information" during training. A key future direction is to use our state-based PRM4RL to generate high-quality, dense rewards to **guide and accelerate the training of a vision-based policy**. Our work provides the algorithmic foundation for this future direction.
>
> ### **Weakness 3**
> > Reviewer's Comment:
> >  "No analysis of how robust Φ is to environment changes."
>
> **Our Response:**
>
> A new experiment has been conducted to analyze the robnustness. We used the **exact same Φ function** to train three separate policies from scratch in three environments **with different physical dynamics.**
> The results are summarized below:
>
> | Training Env | Final Success Rate | Steps to Convergence |
> |:-:|:-:|:-:|
> | Original Env  | 1.00  |  $\approx$ 1.25M     |
> | Object Mass x1.5  | 0.99   |  $\approx$ 1.60M     |
> | Friction Param x1.5 | 1.00  |  $\approx$ 1.10M     |
>
> The policies trained with the same Φ function achieve near-perfect success in all conditions, which demonstrates that **Φ is a robust reward signal across a range of environment conditions,** not overfit to specific dynamics.
>
> ---
>
> ### **Question 1**
> > **Reviewer's Question:**
> > "Please compare to a hand-coded distance-to-goal or potential-based reward to isolate LLM contributions."
>
> **Our Response:**
>
> We introduce two hand-coded baseline in new experiments:
>
> 1. Hand-coded Potential-Based Reward: An expert-designed, potential-based reward requiring significant manual engineering following [1].
>
> 2. Hand-coded Distance-to-Goal (Oracle): The default environment reward function, which acts as a naive hand-coded distance-based signal.
>
> | Method| MW-drawer | MW-window | MW-faucet | MS-PickCube | MS-PullCube |
> |:-:|:--:|:--:|:-:|:-:|:-:|
> | Our Method | 1.00      | 1.00      | 1.00      | 0.99        | 1.00        |
> | Hand-coded potential-based reward  | 1.00      |  0.99     |  1.00     |  1.00       |  1.00       |
> | Hand-coded distance-to-goal reward (Oracle)| 0.75      | 0.51      | 0.26      | 0.19        | 0.13        |
>
> The results show a clear hierarchy: naive Distance-to-Goal rewards fail on complex tasks, while our automated `PRM4RL` consistently **matches the performance of a meticulously hand-engineered potential-based reward.** This powerfully demonstrates that our framework successfully automates expert-level reward design.
>
> Reference: [1] Ng, Andrew Y., Daishi Harada, and Stuart Russell. "Policy invariance under reward transformations: Theory and application to reward shaping." Icml. Vol. 99. 1999.
>
> ### **Question 2**
> > **Reviewer's Question:**
> >  "What is the actual token and wall-clock cost per episode for PRM4RL? How does performance scale to 7B-parameter models?"
>
> **Our Response:**
>
> 1. **Cost Analysis:**
>
>    Our method's cost is a single, upfront expense, not a recurring per-episode cost. The specific numbers for this one-time call are detailed below (Metaworld-pick-place task, GPT-4o).
>
>     | Method      | LLM Calls | Token Cost (Input / Output) | Wall-Clock Cost  |
>     |:-:|:-:|:-:|:-:|
>     | Our Method  | 1  | 875 / 628 Tokens | 9.776 seconds|
>
>     What is crucial to understand is that this "One-Time Generation" represents **a fundamentally new and more efficient LLM Usage Pattern** compared to prior paradigms, as the table below shows, our method achieves the **best performance with the lowest cost**.
>
>
>     | Method| LLM Usage Pattern | Total LLM Calls(Typical 1M-step Training) | Average Success Rate on Metaworld / Maniskill |
>     |:-:|:-:|:-:|:-:|
>     | Our Method  | One-Time  | 1  | 0.99 / 0.98         |
>     | Text2Reward | Iterative, Human-in-the-Loop Refinement | N (N $\geqslant$ 1 cycles)                 | 0.62 / 0.26         |
>     | ELLM        | Continuous, Per-Timestep Planning       | $\approx 1000000$                          | 0.68 / 0.38         |
>
> 2. **Scaling to Smaller Models:**
>    We conducted new experiments with a open-source 8B model (Qwen3-8B). We evaluated both the model's ability to generate a valid function ("Generation Success Rate") and the performance of the policy trained with that function ("Task Success Rate").
>
>    | LLM|  Generation Success Rate| Task Success Rate(if generated) |
>    |:-:|:-:|:-:|
>    | GPT-4o   |1.0 | 1.00 |
>    | Qwen3-8B |0.9 (9/10 runs)| 0.99|
>
>    These results show that our structured prompting makes the generation task feasible even for a much smaller 8B model, which produced high-performing reward functions with high reliability.
>
> ### **Question 3**
> > **Reviewer's Question:**
> >  "Does the same Φ generalize if the task dynamics change (e.g., object mass or friction)?"
>
> **Our Response:**
>
> We respectfully refer you to our **response to `Weakness #3`** above.
>
> ### **Question 4**
> > **Reviewer's Question:**
> >  "How does the method detect or recover from a misleading or suboptimal Φ generated by the LLM?"
>
> **Our Response:**
>
> We interpret a "misleading or suboptimal" generation as a form of LLM hallucination, which is a known and fundamental challenge for all LLM-based methods. Our primary approach is not passive recovery, but **proactive mitigation** through a two-pronged strategy:
>
> 1. **Framework Structure:** By **decomposing a complex, long-horizon problem into a series of simple, manageable subtasks**, we fundamentally reduce the complexity of the code generation and task understanding challenge. This is **empirically validated** by our method's success on multi-stage tasks (e.g., **1.00 success** on `MW-pick-place`) where the monolithic Text2Reward approach fails (**0.00 success**).
>
> 2. **Structured Prompting:** For each simple subtask, our **structured, Pythonic prompt** communicates the task in a format that code-specialized LLMs are best at understanding. By providing a clear, class-based "API" of the environment information (see `Appendix E`), we **minimize ambiguity** and guide the LLM to generate code within this well-defined, familiar structure. **A new ablation experiments** confirms its importance: replacing it with a standard natural language prompt dropped the generation success rate to just 30%, demonstrating our approach is crucial for reliability.
>
>    In practice, we found this two-pronged approach to be highly reliable, leading to the successful generation of valid, high-quality functions for all 15 diverse tasks in our experiments.
>
> * **Limitations and Future Work:** We see great potential in **synergizing our method with iterative refinement systems** like Eureka [1]. PRM4RL could provide **a high-quality "perfect initialization"** for such systems, drastically reducing their search space. We believe this combination is a promising direction for future work.
>
>   References: [1] Ma, Yecheng Jason, et al. "Eureka: Human-level reward design via coding large language models." arXiv preprint arXiv:2310.12931 (2023).

---

> > ### Comment · Reviewer_MTbw · 2025-08-08
> >
> > I appreciate authors for their further feedback. I would be happy to increase my score and encourage authors to put the new results in the revisions.

---

### Note · Authors · 2025-08-15

## FInal Remaks
---

We sincerely **thank the Area Chair and all reviewers** for their time and insightful feedback throughout this constructive review process.
We are grateful for the opportunity to have engaged in a productive discussion, and we are encouraged by the **positive responses and score increases,** which we believe reflect that **we have successfully addressed the reviewers' primary concerns.**

To summarize, our work, PRM4RL, makes two primary contributions to LLM-augmented Reinforcement Learning:
* **A Principled, Theoretically-Grounded Reward Model:** We are the first to introduce and formalize the use of potential-based reward shaping in this domain, providing our LLM-generated rewards with proven policy invariance and convergence guarantees.

* **A Highly Efficient and Synergistic Structure:** We propose a new "Single-Pass Generation" paradigm where one LLM call generates the entire hierarchical setup (plan, tracker, and reward). Our experiments show this is not only orders of magnitude more efficient than prior paradigms but also achieves state-of-the-art performance, especially on complex, long-horizon tasks.

In direct response to the reviewers' excellent suggestions, **we conducted several new experiments** during the rebuttal period. These new results, shared in our responses, provide further empirical evidence of our framework's robustness to changing dynamics, its scalability to smaller open-source models, and its superiority over additional hand-coded baselines.

We are confident that these clarifications and new results demonstrate that PRM4RL is a significant, principled, and practical contribution to the field. Thank you once again for your careful consideration of our work.

---

### Decision · Program_Chairs · 2025-09-17

**Decision:**

Accept (poster)

**Comment:**

This work makes an attempt to develop the reward model in RL by LLMs. The work includes both a theoretical guarantee and an algorithmic structure. All reviewers agree that the work provides significant contributions to the community. I therefore recommend acceptance.